# Testing Calibration in Nearly-Linear Time

**Lunjia Hu**
Harvard University
lunjia@alumni.stanford.edu

**Arun Jambulapati**
University of Michigan
jmblpati@gmail.com

**Kevin Tian**
University of Texas at Austin
kjtian@cs.utexas.edu

**Chutong Yang**
University of Texas at Austin
cyang98@utexas.edu

## Abstract

In the recent literature on machine learning and decision making, *calibration* has emerged as a desirable and widely-studied statistical property of the outputs of binary prediction models. However, the algorithmic aspects of measuring model calibration have remained relatively less well-explored. Motivated by [BGHN23a], which proposed a rigorous framework for measuring distances to calibration, we initiate the algorithmic study of calibration through the lens of property testing. We define the problem of *calibration testing* from samples where given $n$ draws from a distribution $\mathcal{D}$ on (predictions, binary outcomes), our goal is to distinguish between the cases where $\mathcal{D}$ is perfectly calibrated or $\varepsilon$-far from calibration. We make the simple observation that the empirical smooth calibration linear program can be reformulated as an instance of minimum-cost flow on a highly-structured graph, and design an exact dynamic programming-based solver for it which runs in time $O(n \log^2(n))$, and solves the calibration testing problem information-theoretically optimally in the same time. This improves upon state-of-the-art black-box linear program solvers requiring $\Omega(n^\omega)$ time, where $\omega > 2$ is the exponent of matrix multiplication. We also develop algorithms for tolerant variants of our testing problem improving upon black-box linear program solvers, and give sample complexity lower bounds for alternative calibration measures to the one considered in this work. Finally, we present experiments showing the testing problem we define faithfully captures standard notions of calibration, and that our algorithms scale efficiently to accommodate large sample sizes.

## 1 Introduction

Probabilistic predictions are at the heart of modern data science. In domains as wide-ranging as forecasting (e.g. predicting the chance of rain from meteorological data [MW84, Mur98]), medicine (e.g. assessing the likelihood of disease [Doi07]), computer vision (e.g. assigning confidence values for categorizing images [VDDP17]), and more (e.g. speech recognition [AAA+16] and recommender systems [RRSK11]), prediction models have by now become essential components of the decision-making pipeline. Particularly in the context of critical, high-risk use cases, the interpretability of prediction models is therefore paramount in downstream applications. That is, how do we assign meaning to the predictions our model gives us, especially when the model is uncertain?

We focus on perhaps the most ubiquitous form of prediction modeling: binary predictions, represented as tuples $(v, y)$ in $[0, 1] \times \{0, 1\}$ (where the $v$ coordinate is our prediction of the likelihood of an event, and the $y$ coordinate is the observed outcome). We model prediction-outcome pairs in the binary prediction setting by a joint distribution $\mathcal{D}$ over $[0, 1] \times \{0, 1\}$, fixed in the following discussion. In

38th Conference on Neural Information Processing Systems (NeurIPS 2024).

this context, *calibration* of a predictor has emerged as a basic desideratum. A prediction-outcome distribution $\mathcal{D}$ is said to be calibrated if

$$\mathbb{E}_{(v,y)\sim\mathcal{D}}[y \mid v = t] = t \text{ for all } t \in [0,1]. \tag{1}$$

That is, calibration asks that the outcome is $1$ exactly $60\%$ of the time, when the model returns a prediction $v = 0.6$. While calibration (or approximate variants thereof) is a relatively weak requirement on a meaningful predictor, as it can be achieved by simple models,[1] it can still be significantly violated in practice. For example, interest in calibration in the machine learning community was spurred by [GPSW17], which observed that many modern deep learning models are far from calibrated. Moreover, variants of calibration have been shown to have strong postprocessing properties for fairness constraints and loss minimization [HKRR18, DKR+21, GKR+22], which has garnered renewed interest in calibration by the theoretical computer science and statistics communities.

The question of measuring the calibration of a distribution is subtle; even a calibrated distribution incurs measurement error due to sampling. For example, consider the *expected calibration error*, used in e.g. [NCH15, GPSW17, MDR+21a, RT21b] as a ground-truth measure of calibration:

$$\mathsf{ECE}(\mathcal{D}) := \mathbb{E}_{(v,y)\sim\mathcal{D}}\left[\left|\mathbb{E}_{(v',y)\sim\mathcal{D}}\left[y \mid v' = v\right] - v\right|\right].$$

Unfortunately, the empirical ECE is typically meaningless; if the marginal density of $v$ is continuous, we will almost surely only observe a single sample with each $v$ value. Further, [KF08] observed that ECE is discontinuous in $v$. In practice, binned variants of ECE are often used as a proxy, where a range of $v$ is lumped together in the conditioning event. However, hyperparameter choices (e.g. the number of bins) can significantly affect the quality of binned ECE variants as a distance measure [KLM19, NDZ+19, MDR+21a].[2] Moreover, as we explore in this paper, binned calibration measures inherently suffer from larger sample complexity-to-accuracy tradeoffs, and are less faithful to ground truth calibration notions in experiments than the calibration measures we consider.

Recently, [BGHN23a] undertook a systematic study of various measures of distance to calibration proposed in the literature. They proposed information-theoretic tractability in the *prediction-only access (PA) model*, where the calibration measure definition can only depend on the joint prediction-outcome distribution (rather than the features of training examples),[3] as a desirable criterion for calibration measures. Correspondingly, [BGHN23a] introduced Definition 1 as a ground-truth notion for measuring calibration in the PA model, which we also adopt in this work.[4]

**Definition 1** (Lower distance to calibration). *Let $\mathcal{D}$ be a distribution over $[0,1] \times \{0,1\}$. The* lower distance to calibration (LDTC) *of $\mathcal{D}$, denoted $\underline{\mathsf{dCE}}(\mathcal{D})$, is defined by*

$$\underline{\mathsf{dCE}}(\mathcal{D}) := \inf_{\Pi \in \mathrm{ext}(\mathcal{D})} \mathbb{E}_{(u,v,y)\sim\Pi} |u - v|,$$

*where* $\mathrm{ext}(\mathcal{D})$ *is all distributions $\Pi$ over $(u,v,y) \in [0,1] \times [0,1] \times \{0,1\}$ satisfying the following.*

- *The marginal distribution of $(v,y)$ is $\mathcal{D}$.*

- *The marginal distribution $(u,y)$ is perfectly calibrated, i.e. $\mathbb{E}_\Pi[y|u] = u$.*

Definition 1 has various beneficial aspects: it is convex in $v$, computable in the PA model, and (as shown by [BGHN23a]) polynomially-related to various other calibration measures, including some which require feature access, e.g. the *distance to calibration* (DTC, Eq. (1), [BGHN23a]). Roughly, the DTC of a distribution is the tightest lower bound on the $\ell_1$ distance between $v$ and any calibrated function of the features, *after taking features into account*. The LDTC is the analog of this feature-aware measure of calibration when limited to the PA model, so it does not depend on features. We focus on Definition 1 as our ground-truth measure in the remainder of the paper.

---

[1]The predictor which ignores features and always return the population mean is calibrated, for example.

[2]For example, [NDZ+19] observed that, in their words, "dramatic differences in bin sensitivity" can occur "depending on properties of the (distribution) at hand," a sentiment echoed by Section 5 of [MDR+21a].

[3]This access model is practically desirable because it abstracts away the feature space, which can lead to significant memory savings when our goal is only to test the calibration of model predictions. Moreover, this matches conventions in the machine learning literature, e.g. loss functions are typically defined in the PA model.

[4]We note that [BGHN23a] introduced an *upper distance to calibration*, also defined in the PA model, which they showed is quadratically-related to the $\underline{\mathsf{dCE}}$ in Definition 1. However, the upper distance does not satisfy basic properties such as continuity, making it less amenable to estimation and algorithm design.

## 1.1 Our results

We initiate the algorithmic study of the *calibration testing* problem, defined as follows.

**Definition 2** (Calibration testing). *Let $\varepsilon \in [0, 1]$. We say algorithm $\mathcal{A}$ solves the $\varepsilon$-calibration testing problem with $n$ samples, if given $n$ i.i.d. draws from a distribution $\mathcal{D}$ over $[0, 1] \times \{0, 1\}$, $\mathcal{A}$ returns either "yes" or "no" and satisfies the following with probability $\geq \frac{2}{3}$.*[5]

- *$\mathcal{A}$ returns "no" if $\underline{\mathsf{dCE}}(\mathcal{D}) \geq \varepsilon$.*
- *$\mathcal{A}$ returns "yes" if $\underline{\mathsf{dCE}}(\mathcal{D}) = 0$.*

*In this case, we also call $\mathcal{A}$ an $\varepsilon$-calibration tester.*

To our knowledge, we are the first to formalize the calibration testing problem in Definition 2, which is natural from the perspective of *property testing*, an influential paradigm in statistical learning [Ron08, Ron09, Gol17]. In particular, there is an $\varepsilon_n = \Theta(n^{-1/2})$ so that it is information-theoretically impossible to solve the $\varepsilon_n$-calibration testing problem from $n$ samples (see Lemma 5), so a variant of Definition 2 with an exact distinguishing threshold between "calibrated/uncalibrated" is not tractable. Hence, Definition 2 only requires distinguishing distributions $\mathcal{D}$ which are "clearly uncalibrated" (parameterized by a threshold $\varepsilon$) from those which are perfectly calibrated.

We note that a variant of Definition 2 where $\underline{\mathsf{dCE}}$ is replaced by variants of the ECE was recently proposed by [LHHD23]. However, due to the aforementioned discontinuity and binning choice issues which plague the ECE, [LHHD23] posed as an explicit open question whether an alternative calibration metric makes for a more appropriate calibration testing problem, motivating our Definition 2. Indeed, Proposition 9 of [LHHD23] shows that without smoothness assumptions on the data distribution, it is impossible to solve the ECE calibration testing problem from finite samples.[6]

Our first algorithmic contribution is a nearly-linear time algorithm for calibration testing.

**Theorem 1.** *Let $n \in \mathbb{N}$, and $\varepsilon = \Omega(\varepsilon_n)$, where $\varepsilon_n = \Theta(n^{-1/2})$ is minimal such that it is information-theoretically possible to solve the $\varepsilon_n$-calibration testing problem (Definition 2) with $n$ samples. There is an algorithm which solves the $\varepsilon$-calibration testing problem with $n$ samples, running in time $O(n \log^2(n))$.*

The lower bound on the acceptable range of $\varepsilon_n$ in Theorem 1 is well-known, and recalled in Lemma 5 for completeness. Our main contribution is to prove the upper bound (i.e., achieving $O(\varepsilon_n)$-calibration testing) in Theorem 1 by designing a new algorithm for computing $\mathsf{smCE}(\widehat{\mathcal{D}}_n)$, the *smooth calibration error* (Definition 3), an alternative calibration measure, of an empirical distribution $\widehat{\mathcal{D}}_n$.

**Definition 3** (Smooth calibration error). *Let $W$ be the set of Lipschitz functions $w : [0, 1] \to [-1, 1]$. The* smooth calibration error *of distribution $\mathcal{D}$ over $[0, 1] \times \{0, 1\}$, denoted $\mathsf{smCE}(\mathcal{D})$, is*

$$\mathsf{smCE}(\mathcal{D}) = \sup_{w \in W} \left| \mathbb{E}_{(v,y) \sim \mathcal{D}}[(y - v)w(v)] \right|.$$

It was shown in [BGHN23a] that $\mathsf{smCE}(\mathcal{D})$ is a constant-factor approximation to $\underline{\mathsf{dCE}}(\mathcal{D})$ for all $\mathcal{D}$ on $[0, 1] \times \{0, 1\}$ (see Lemma 4). Additionally, the empirical $\mathsf{smCE}$ admits a representation as a linear program with an $O(n) \times O(n)$-sized constraint matrix encoding Lipschitz constraints.[7] Thus, [BGHN23a] proposed a simple procedure for estimating $\mathsf{smCE}(\mathcal{D})$: draw $n$ samples from $\mathcal{D}$, and solve the associated linear program on the empirical distribution. While there have been significant recent runtime advances in the linear programming literature [LS14, CLS21, vdBLSS20, vdBLL+21], all state-of-the-art black-box linear programming algorithms solve linear systems involving the constraint matrix, which takes $\Omega(n^\omega)$ time, where $\omega > 2.371$ [WXXZ23] is the current exponent of

---

[5]As is standard in property testing problems, the success probability of either a calibration tester or a tolerant calibration tester can be boosted to $1 - \delta$ for any $\delta \in (0, 1)$ at a $k = O(\log(\frac{1}{\delta}))$ overhead in the sample complexity. This is because we can independently call $k$ copies of the tester and output the majority vote, which succeeds with probability $\geq 1 - \delta$ by Chernoff bounds, so we focus on $\delta = \frac{1}{3}$.

[6]The work [LHHD23] considered $k$-class prediction tasks, extending our focus on binary classification ($k = 2$), which we believe is an exciting future direction. However, their Proposition 9 holds even when $k = 2$.

[7]Formally, the number of constraints in the $\mathsf{smCE}$ linear program is $O(n^2)$, but we show that in the hard-constrained setting, requiring that "adjacent" constraints are met suffices (see Lemma 1).

matrix multiplication. Even under the best-possible assumption that $\omega = 2$, the strategy of exactly solving a linear program represents an $\Omega(n^2)$ quadratic runtime barrier for calibration testing.

We bypass this barrier by noting that the smCE linear program is highly-structured, and can be reformulated as minimum-cost flow on a planar graph. We believe this observation is already independently interesting, as it opens the door to using powerful software packages designed for efficiently solving flow problems to measure calibration in practice. Moreover, using recent theoretical breakthroughs in graph algorithms [DGG$^+$22, CKL$^+$22] as a black box, this observation readily implies an $O(n \cdot \text{polylog}(n))$-time algorithm for solving the smooth calibration linear program.

However, these aforementioned algorithms are quite complicated, and implementations in practice are not available, leaving their relevance to empirical calibration testing unclear at the moment. Motivated by this, in Section 2 we develop a custom solver for the minimum-cost flow reformulation of empirical smooth calibration, based on dynamic programming. Our theoretical runtime improvement upon [DGG$^+$22, CKL$^+$22] is by at least a large polylogarithmic factor, and moreover our algorithm is simple enough to implement in practice, where it attains faster runtimes than general-purpose commercial solvers on moderate or large dataset sizes, evaluated in Section 3.

We further define a *tolerant* variant of Definition 2 (see Definition 4), where we allow for error thresholds in both the "yes" and "no" cases; "yes" is the required answer when $\underline{\mathsf{dCE}}(\mathcal{D}) \leq \varepsilon_2$, and "no" is required when $\underline{\mathsf{dCE}}(\mathcal{D}) \geq \varepsilon_1$. Our algorithm in Theorem 1 continues to serve as an efficient tolerant calibration tester when $\varepsilon_1 \geq 4\varepsilon_2$, formally stated in Theorem 3. This constant-factor loss comes from a similar loss in the relationship between smCE and $\underline{\mathsf{dCE}}$, see Lemma 4. We make the observation that a constant factor loss in the tolerant testing parameters is inherent following this strategy, via a lower bound in Lemma 13. Thus, even given infinite samples, computing the smooth calibration error cannot solve tolerant calibration testing all the way down to the information-theoretic threshold $\varepsilon_1 \geq \varepsilon_2$. To develop an improved tolerant calibration tester, we directly show how to approximate the LDTC of an empirical distribution, our second main algorithmic contribution.

**Theorem 2** (Informal, see Theorem 4, Corollary 3). *Let $n \in \mathbb{N}$, and let $\varepsilon_1 - \varepsilon_2 = \Omega(\varepsilon_n)$, where $\varepsilon_n = \Theta(n^{-1/2})$ is minimal such that it is information-theoretically possible to solve the $\varepsilon_n$-calibration testing problem (Definition 4) with $n$ samples. There is an algorithm which solves the $(\varepsilon_1, \varepsilon_2)$-tolerant calibration testing problem with $n$ samples, running in time $O(\frac{n \log(n)}{(\varepsilon_1 - \varepsilon_2)^2}) = O(n^2 \log(n))$.*

While Theorem 2 is slower than Theorem 1, it directly approximates the LDTC, making it applicable to tolerant calibration testing. We mention that state-of-the-art black-box linear programming based solvers, while still applicable to (a discretizeation of) the empirical LDTC, require $\Omega(n^{2.5})$ time [vdBLL$^+$21], even if $\omega = 2$. This is because the constraint matrix for the $\varepsilon$-approximate empirical LDTC linear program has dimensions $O(\frac{n}{\varepsilon}) \times O(n)$, resulting in an $\approx \frac{1}{\varepsilon} = \Omega(\sqrt{n})$ overhead in the dimension of the decision variable. We prove Theorem 2 in Appendix C, where we use recent advances in minimax optimization [JT23] and a custom combinatorial rounding procedure to develop a faster algorithm, improving state-of-the-art linear programming runtimes by an $\Omega(\sqrt{n})$ factor.

In Appendix D, we complement our algorithmic results with lower bounds (Theorems 5, 6) on the sample complexity required to solve variants of the testing problem in Definition 2, when $\underline{\mathsf{dCE}}$ is replaced with different calibration measures. For several widely-used distances in the machine learning literature, including binned and convolved variants of ECE [NCH15, BN23], we show that $\widetilde{\Omega}(\varepsilon^{-2.5})$ samples are required to the associated $\varepsilon$-calibration testing problem. This demonstrates a statistical advantage of our focus on $\underline{\mathsf{dCE}}$ as our ground-truth notion for calibration testing.

We corroborate our theoretical findings with experimental evidence on real and synthetic data in Section 3. First, on a simple Bernoulli example, we show that $\underline{\mathsf{dCE}}$ and smCE testers are more reliable measures of calibration than a recently-proposed binned ECE variant. We then apply our smCE tester to postprocessed neural network predictions to test their calibration levels, validating against the findings in [GPSW17]. Finally, we implement our method from Theorem 1 on our Bernoulli dataset, showing that it scales to high dimensions and runs faster than both a linear program solver from CVXPY for computing the empirical smCE, as well as a commercial minimum-cost flow solver from Gurobi Optimization (combined with our reformulation in Lemma 2).[8]

---

[8]Our code is included in the supplementary material.

## 1.2 Our techniques

Theorems 1 and 2 follow from designing custom algorithms for approximating empirical linear programs associated with the smCE and dCE of a sampled dataset $\widehat{\mathcal{D}}_n := \{(v_i, y_i)\}_{i \in [n]} \sim_{\text{i.i.d.}} \mathcal{D}$. In both cases, generalization bounds from [BGHN23a] show it suffices to approximate the value of the empirical calibration measures to error $\varepsilon = \Omega(n^{-1/2})$, though our solver in Theorem 1 will be exact.

We begin by explaining our strategy for estimating $\text{smCE}(\widehat{\mathcal{D}}_n)$ (Definition 3). By definition, the smooth calibration error of $\widehat{\mathcal{D}}_n$ can be formulated as a linear program,

$$\min_{x \in [-1,1]^n} \frac{1}{n} \sum_{i \in [n]} x_i(v_i - y_i), \text{ where } |x_i - x_j| \le |v_i - v_j| \text{ for all } (i,j) \in [n] \times [n]. \quad (2)$$

Here, $x_i \in [-1, 1]$ corresponds to the weight on $v_i$, and there are $2\binom{n}{2}$ constraints on the decision variable $x$, each of which corresponds to a Lipschitz constraint. We make the simple observation that every Lipschitz inequality constraint can be replaced by two constraints of the form $x_i - x_j \le |v_j - v_i|$ (with $i, j$ swapped). Moreover, the box constraints $x \in [-1,1]^n$ can be handled by introducing a dummy variable $x_{n+1}$ and writing $\max(x_i - x_{n+1}, x_{n+1} - x_i) \le 1$, after penalizing $x_{n+1}$ appropriately in the objective. Notably, this substitution makes every constraint the difference of two decision variables, which is enforceable using the edge-vertex incidence matrix of a graph. Finally, the triangle inequality implies that we only need to enforce Lipschitz constraints in (2) corresponding to adjacent $i, j$. After making these simplifications, the result is the dual of a minimum-cost flow problem on a graph which is the union of a star and a path; this argument is carried out in Lemma 2.

Because of the sequential structure of the induced graph, we show in Appendix B.2 that a dynamic programming-based approach, which maintains the minimum-cost flow value after committing to the first $i < n$ flow variables in the graph recursively, succeeds in computing the value (2). To implement each iteration of our dynamic program in polylogarithmic time, we rely on a generalization of the classical segment tree data structure that we develop in Appendix B.3; combining gives Theorem 1.

On the other hand, the linear program corresponding to the empirical dCE is more complex (with two types of constraints), and to our knowledge lacks the graphical structure to be compatible with the aforementioned approach. Moreover, it is not obvious how to use first-order methods, an alternative linear programming framework suitable when only approximate answers are needed, to solve this problem more quickly. This is because the empirical dCE linear program enforces hard constraints to a set that is difficult to project to under standard distance metrics. To develop our faster algorithm in Theorem 2, we instead follow an "augmented Lagrangian" method where we lift the constraints directly into the objective as a soft-constrained penalty term. To prove correctness of this lifting, we follow a line of results in combinatorial optimization [She13, JST19]. These works develop a "proof-by-rounding algorithm" framework to show that the hard-constrained and soft-constrained linear programs have equal values, summarized in Appendix C.1 (see Lemma 14).

To use this augmented Lagrangian framework, it remains to develop an appropriate rounding algorithm to the feasible polytope for the empirical dCE linear program, which enforces two types of constraints: marginal satisfaction of $(v, y)$, and calibration of $(u, y)$ (using notation from Definition 1). In Appendix C.3, we design a two-step rounding procedure, which first fixes the marginals on the $(v, y)$ coordinates, and then calibrates the $u$ coordinates without affecting any $(v, y)$ marginal.

## 1.3 Related work

The calibration performance of deep neural networks has been studied extensively in the literature (e.g. [GPSW17, MDR$^+$21b, Rt21a, BGHN23b]). Measuring the calibration error in a meaningful way can be challenging, especially when the predictions are not naturally discretized (e.g. in neural networks). Recently, [BGHN23a] addresses this challenge using the *distance to calibration* as a central notion. They consider a calibration measure to be *consistent* if it is polynomially-related to the distance to calibration. Consistent calibration measures include the smooth calibration error [KF04], Laplace kernel calibration error [KSJ18], interval calibration error [BGHN23a], and convolved ECE [BN23].[9]

---

[9]The calibration measure we call the *convolved ECE* in our work was originally called the *smooth ECE* in [BN23]. We change the name slightly to reduce overlap with the smooth calibration error (Definition 3), a central object throughout the paper.

On the algorithmic front, substantial observations were made by [BGHN23a] on linear programming characterizations of calibration measures such as the LDTC and smooth calibration. While there have been significant advances on the runtime frontier of linear programming solvers, current runtimes for handling an $n \times d$ linear program constraint matrix with $n \geq d$ remain $\Omega(\min(nd + d^{2.5}, n^\omega))$ [CLS21, vdBLL$^+$21, JSWZ21]. Our constraint matrix is roughly-square and highly-sparse, so it is plausible that e.g. the recent research on sparse linear system solvers [PV21, Nie22] could apply to the relevant Newton's method subproblems and improve upon these rates. Moreover, while efficient estimation algorithms have been proposed by [BGHN23a] for (surrogate) interval calibration error and by [BN23] for convolved ECE, these algorithms require suboptimal sample complexity for solving our testing task in Definition 2 (see Appendix D). To compute their respective distances to error $\varepsilon$ from samples, these algorithms require $\Omega(\varepsilon^{-5})$ and $\Omega(\varepsilon^{-3})$ time. As comparison, under this parameterization Theorems 1 and 2 require $\widetilde{O}(\varepsilon^{-2})$ and $\widetilde{O}(\varepsilon^{-4})$ time, but can solve stronger testing problems with the same sample complexity, experimentally validated in Section 3.

**Notation.**  Throughout, $\mathcal{D}$ denotes a distribution over $[0, 1] \times \{0, 1\}$. When $\mathcal{D}$ is clear from context, we let $\widehat{\mathcal{D}}_n = \{(v_i, y_i)\}_{i \in [n]}$ denote a dataset of $n$ independent samples from $\mathcal{D}$ and, in a slight abuse of notation, the distribution with probability $\frac{1}{n}$ for each $(v_i, y_i)$. We say d is a *calibration measure* if it takes distributions on $[0, 1] \times \{0, 1\}$ to the nonnegative reals $\mathbb{R}_{\geq 0}$, so dCE (Definition 1) and smCE (Definition 3) are both calibration measures. We use $\widetilde{O}$ and $\widetilde{\Omega}$ to hide polylogarithmic factors in the argument. We denote $[n] := \{i \in \mathbb{N} \mid i \leq n\}$. We denote matrices in boldface throughout. For any $\mathbf{A} \in \mathbb{R}^{m \times n}$, we refer to its $i^{\text{th}}$ row by $\mathbf{A}_{i:}$ and its $j^{\text{th}}$ column by $\mathbf{A}_{:j}$. For a set $S$ identified with rows of a matrix $\mathbf{A}$, we let $\mathbf{A}_{s:}$ denote the row indexed by $s \in S$, and use similar notation for columns. For a directed graph $G = (V, E)$, we define its edge-vertex incidence matrix $\mathbf{B} \in \{-1, 0, 1\}^{E \times V}$ which has a row corresponding to each $e = (u, v) \in E$ with $\mathbf{B}_{eu} = 1$ and $\mathbf{B}_{ev} = -1$. When $G$ is undirected, we similarly define $\mathbf{B} \in \{-1, 0, 1\}^{E \times V}$ with arbitrary edge orientations.

## 2   Smooth calibration

In this section, we overview our main result on approximating the smooth calibration of a distribution on $[0, 1] \times \{0, 1\}$, deferring some aspects of the proof to Appendix B. We first show that the linear program corresponding to the smooth calibration of an empirical distribution can be reformulated as an instance of minimum-cost flow on a highly-structured graph. We then explain our dynamic programming approach to solving this minimum-cost flow problem and state a runtime guarantee. Finally, we give our main result on near-linear time calibration testing, Theorem 1.

Throughout this section, we fix a dataset under consideration, $\widehat{\mathcal{D}}_n := \{(v_i, y_i)\}_{i \in [n]} \subset [0, 1] \times \{0, 1\}$, and the corresponding empirical distribution (which, in an abuse of notation, we also denote $\widehat{\mathcal{D}}_n$), i.e. we use $(v, y) \sim \widehat{\mathcal{D}}_n$ to mean that $(v, y) = (v_i, y_i)$ with probability $\frac{1}{n}$ for each $i \in [n]$. We also assume without loss of generality that the $\{v_i\}_{i \in [n]}$ are in sorted order, so $0 \leq v_1 \leq \ldots \leq v_n \leq 1$. Recalling Definition 3, the associated empirical smooth calibration linear program is

$$\mathsf{smCE}(\widehat{\mathcal{D}}_n) := \max_{x \in [-1, 1]^n} b^\top x,$$

$$\text{where } |x_i - x_j| \leq v_j - v_i \text{ for all } (i, j) \in [n] \times [n] \text{ with } i < j, \tag{3}$$

$$\text{and } b_i := \frac{1}{n}(y_i - v_i) \text{ for all } i \in [n].$$

We first make a simplifying observation, which shows that it suffices to replace the Lipschitz constraints in (3) with only the Lipschitz constraints corresponding to adjacent indices $(i, j)$.

**Lemma 1.** *If $x, v \in \mathbb{R}^n$, where $v$ has monotonically nondecreasing coordinates, and $|x_i - x_{i+1}| \leq v_{i+1} - v_i$ for all $i \in [n-1]$, then $|x_i - x_j| \leq v_j - v_i$ for all $(i, j) \in [n] \times [n]$ with $i < j$.*

We now reformulate (3) as a (variant of a) minimum-cost flow problem.

**Lemma 2.** *Consider an instance of (3). Let $G = (V, E)$ be an undirected graph on $n + 1$ vertices labeled by $V := [n + 1]$, and with $2n - 1$ directed edges $E$ defined and with edge costs as follows.*

- *For all $i \in [n-1]$, there is an edge between vertices $(i, i + 1)$ with edge cost $v_{i+1} - v_i$.*

- *For all $i \in [n]$, there is an edge between vertices $(i, n+1)$ with edge cost 1.*

*Let $c \in \mathbb{R}^E$ be the vector of all edge costs, let $d \in \mathbb{R}^{n+1}$ be the demand vector which concatenates $-b$ in (3) with a last coordinate set to $\sum_{i \in [n]} b_i$, and let $\mathbf{B} \in \{-1, 0, 1\}^{E \times V}$ be the edge-vertex incidence matrix of $G$. Then the problem*

$$\min_{\substack{f \in \mathbb{R}^E \\ \mathbf{B}^\top f = d}} c^\top |f| := \sum_{e \in E} c_e |f_e| \tag{4}$$

*has the same value as the empirical smooth calibration linear program (3).*

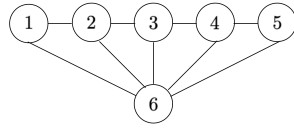

Figure 1: Example graph $G$ for $n = 5$ with $n + 1 = 6$ vertices and $2n - 1 = 9$ edges.

*Proof.* By Lemma 1, solving (3) is equivalent to solving

$$\min_{x \in [-1,1]^n} -b^\top x, \text{ where } |x_i - x_{i+1}| \le v_{i+1} - v_i \text{ for all } i \in [n-1], \tag{5}$$

We create a dummy variable $x_{n+1}$, and rewrite (5) as

$$\min_{x \in \mathbb{R}^{n+1}} \sum_{i \in [n]} -b_i(x_i - x_{n+1}), \text{ where } |x_i - x_{i+1}| \le v_{i+1} - v_i \text{ for all } i \in [n-1],$$
$$\text{and } -1 \le x_i - x_{n+1} \le 1 \text{ for all } i \in [n]. \tag{6}$$

Next, consider a directed graph $\widetilde{G} = (V, \widetilde{E})$ with $4n - 2$ edges which duplicate the undirected edges described in the lemma statement in both directions. Let $\widetilde{\mathbf{B}} \in \{-1, 0, 1\}^{\widetilde{E} \times V}$ be the edge-vertex incidence matrix of $\widetilde{G}$, and let $\tilde{c} \in \mathbb{R}^{\widetilde{E}}$ be the edge cost vector so that both edges in $\widetilde{E}$ corresponding to $e \in E$ have the same cost $c_e$. Then (6) is equivalent to the linear program $\max_{x \in \mathbb{R}^{n+1}} d^\top x$ such that $\widetilde{\mathbf{B}} x \le c$, where $d$ is described as in the lemma statement. The dual of this linear program is

$$\min_{\substack{f \in \mathbb{R}^{\widetilde{E}}_{\ge 0} \\ \widetilde{\mathbf{B}}^\top f = d}} \tilde{c}^\top f, \tag{7}$$

a minimum-cost flow problem on $\widetilde{G}$. In particular, based on the way we defined $\widetilde{\mathbf{B}}$, we can check that $\widetilde{\mathbf{B}}^\top f$ encodes the net flow at each vertex of $\widetilde{G}$, which is set according to the demand vector $d$ in the above optimization problem. Next, for each pair of directed edges $(e', e'')$ in $\widetilde{G}$ corresponding to some $e \in E$, note that an optimal solution to (7) will only put nonzero flow on one of $e'$ or $e''$, else we can achieve a smaller cost by canceling out redundant flow. Therefore, we can collapse each pair of directed edges into a single undirected edge, where we allow the flow variable $f$ to be negative but charge its magnitude $|f|$ in cost, proving equivalence of (7) and (4) as claimed. □

We believe this observation in Lemma 2 is already interesting, as it lets us to use specialized graph algorithms to achieve faster runtimes in both theory and practice for solving (3). By using the special structure of the graph (the union of a star and path), we show in Appendix B that we can develop a more efficient custom algorithm for this problem. Specifically, we show how to replace the constrained problem (4) with an unconstrained problem on only the path edges, of the form

$$\min_{f \in \mathbb{R}^{n-1}} A(f) := |d_1 + f_1| + |d_n - f_{n-1}| + \sum_{i \in [n-2]} |f_i - f_{i+1} - d_{i+1}| + \sum_{i \in [n-1]} c_i |f_i|. \tag{8}$$

We prove the following result in Appendix B.2.

**Proposition 1.** *There is an algorithm which computes a minimizer $f \in \mathbb{R}^{n-1}$ to $A$ in (8), as well as the minimizing value $A(f)$, in time $O(n \log^2(n))$.*

Our algorithm for establishing Proposition 1 is based on dynamic programming, and recursively represents partial solutions to $A$ as a piecewise-linear function. We implement updates to this representation via a *segment tree* data structure in polylogarithmic time, giving our overall solution.

**Corollary 1.** *There is an algorithm which computes the value of* (3) *in time* $O(n \log^2(n))$.

*Proof.* This is immediate from Lemma 2, the equivalence between the constrained problem (4) and the unconstrained problem (8) established in Appendix B.2, and Proposition 1. □

We now describe how to build upon Corollary 1 to give an algorithm for proving Theorem 1, using a result from [BGHN23a] which bounds how well the smooth calibration of an empirical distribution approximates the smooth calibration of the population.

**Lemma 3** (Corollary 9.9, [BGHN23a]). *For* $\varepsilon \in (0, 1)$, *there is an* $n = O(\frac{1}{\varepsilon^2})$ *such that if* $\widehat{\mathcal{D}}_n$ *is the empirical distribution over* $n$ *i.i.d. draws from* $\mathcal{D}$, *with probability* $\geq \frac{2}{3}$, $|\mathsf{smCE}(\mathcal{D}) - \mathsf{smCE}(\widehat{\mathcal{D}}_n)| \leq \varepsilon$.

Further, we recall the smooth calibration error is constant-factor related to the LDTC.

**Lemma 4** (Theorem 7.3, [BGHN23a]). *For any distribution* $\mathcal{D}$ *over* $[0, 1] \times \{0, 1\}$, *we have* $\frac{1}{2}\underline{\mathsf{dCE}}(\mathcal{D}) \leq \mathsf{smCE}(\mathcal{D}) \leq 2\underline{\mathsf{dCE}}(\mathcal{D})$.

*Proof of Theorem 1.* We take $n = O(\frac{1}{\varepsilon^2})$ samples so Lemma 3 ensures $|\mathsf{smCE}(\mathcal{D}) - \mathsf{smCE}(\widehat{\mathcal{D}}_n)| \leq \frac{\varepsilon}{2}$ with probability $\geq \frac{2}{3}$. We then compute $\beta = \mathsf{smCE}(\widehat{\mathcal{D}}_n)$ using Corollary 1, and return "yes" iff $\beta \leq \frac{\varepsilon}{4}$, which distinguishes between the two cases in Definition 2 via Lemma 4. □

## 3 Experiments

In this section, we present experiments on synthetic data and CIFAR-100 supporting our argument that $\underline{\mathsf{dCE}}$ and $\mathsf{smCE}$ are reliable measures of calibration for use in defining a testing problem. We then evaluate our custom algorithms from Section 2 and Appendix B, showing promising results on their runtimes outperforming standard packages for linear programming and minimum-cost flow. The experiments in the first and third part of this section are run on a 2018 laptop with 2.2 GHz 6-Core Intel Core i7 processor. The experiments in the second part are run on a cluster using 2x AMD EPYC 7763 64-Core Processor and a single NVIDIA A100 PCIE 40GB.

**Synthetic dataset.** In our first experiment, we considered the ability of $\varepsilon$-d-testers (Definition 5) to detect the miscalibration of a synthetic dataset, for various levels of $\varepsilon \in \{0.01, 0.03, 0.05, 0.07, 0.1\}$, and various choices of $\mathsf{d} \in \{\mathsf{smCE}, \underline{\mathsf{dCE}}, \mathsf{ConvECE}\}$.[10] The synthetic dataset we used is $n$ independent draws from $\mathcal{D}$, where a draw $(v, y) \sim \mathcal{D}$ first draws $v \sim_{\text{unif.}} [0, 1 - \varepsilon^\star]$, and $y \sim \mathsf{Bern}(v + \varepsilon^\star)$, for $\varepsilon^\star := 0.01$.[11] Note that $\underline{\mathsf{dCE}}(\mathcal{D}) = \varepsilon^\star = 0.01$, by the proof in Lemma 13. In Table 1, where the columns index $n$ (the number of samples), for each choice of d we report the smallest value of $\varepsilon$ such that a majority of 100 runs of an $\varepsilon$-d-tester report "yes." For $\mathsf{d} = \mathsf{ConvECE}$, we implemented our tester by running code in [BN23] to compute ConvECE and thresholding at $\frac{\varepsilon}{2}$. For $\mathsf{d} \in \{\mathsf{smCE}, \underline{\mathsf{dCE}}\}$, we used the standard linear program solver from CVXPY [DB16, AVDB18] and again thresholded at $\frac{\varepsilon}{2}$. We remark that the CVXPY solver, when run on the $\underline{\mathsf{dCE}}$ linear program, fails to produce stable results for $n > 2^9$ due to the size of the constraint matrix. As seen from Table 1, both $\mathsf{smCE}$ and $\underline{\mathsf{dCE}}$ testers are more reliable estimators of the ground truth calibration error $\varepsilon^\star$ than ConvECE.

Table 1: Calibration testing thresholds (smallest passed on half of 100 runs).

| $n$ | $2^6 + 1$ | $2^7 + 1$ | $2^8 + 1$ | $2^9 + 1$ | $2^{10} + 1$ | $2^{11} + 1$ |
|---|---|---|---|---|---|---|
| smCE | 0.07 | 0.05 | 0.03 | 0.03 | 0.01 | 0.01 |
| $\underline{\mathsf{dCE}}$ | 0.03 | 0.01 | 0.01 | | | |
| cECE | 0.1 | 0.1 | 0.07 | 0.07 | 0.05 | 0.03 |
| Ground Truth | 0.01 | 0.01 | 0.01 | 0.01 | 0.01 | 0.01 |

---

[10]We implemented ConvECE using code from [BN23], which automatically conducts a search for $\sigma$.

[11]This is a slight variation on the synthetic dataset used in [BGHN23a].

Table 2: Empirical smCE on postprocessed DenseNet40 predictions (median over 20 runs)

| $\mathcal{D}$ | $\mathcal{D}_{\text{base}}$ | $\mathcal{D}_{\text{iso}}$ | $\mathcal{D}_{\text{temp}}$ |
|---|---|---|---|
| Empirical smCE | 0.2269 | 0.2150 | 0.1542 |

In Figure 2, we plot the median error with error bars for each calibration measure, where the $x$ axis denotes $\log_2(n-1)$, and results are reported over 100 runs.

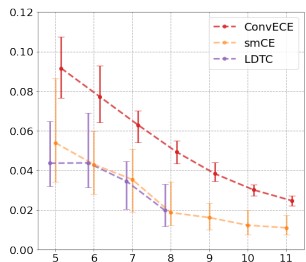

Figure 2: The 25% quantile, median, and 75% quantile (over 100 runs) for smCE, dCE and cECE respectively. The $x$-axis is for dataset with size $2^x + 1$.

**Postprocessed neural networks.** In [GPSW17], which observed modern deep neural networks may be very miscalibrated, various strategies were proposed for postprocessing network predictions to calibrate them. We evaluate two of these strategies using our testing algorithms. We trained a DenseNet40 model [HLvdMW17] on the CIFAR-100 dataset [Kri09], producing a distribution $\mathcal{D}_{\text{base}}$, where a draw $(v, y) \sim \mathcal{D}_{\text{base}}$ selects a random example from the test dataset, sets $y$ to be its label, and $v$ to be the prediction of the neural network. We also learned calibrating postprocessing functions $f_{\text{iso}}$ and $f_{\text{temp}}$ from the training dataset, the former via isotonic regression and the latter via temperature scaling. These induce (ideally, calibrated) distributions $\mathcal{D}_{\text{iso}}$, $\mathcal{D}_{\text{temp}}$, where a draw from $\mathcal{D}_{\text{iso}}$ samples $(v, y) \sim \mathcal{D}_{\text{base}}$ and returns $(f_{\text{iso}}(v), y)$, and $\mathcal{D}_{\text{temp}}$ is defined analogously. The neural network and postprocessing functions were all trained by adapting code from [GPSW17].

We computed the median smooth calibration error of 20 runs of the following experiment. In each run, for each $\mathcal{D} \in \{\mathcal{D}_{\text{base}}, \mathcal{D}_{\text{iso}}, \mathcal{D}_{\text{temp}}\}$, we drew 256 random examples from $\mathcal{D}$, and computed the average smooth calibration error smCE of the empirical dataset using a linear program solver from CVXPY. We report our findings in Table 2. We also compared computing smCE using the CVXPY solver and a commercial minimum-cost flow solver from Gurobi Optimization [Opt23] (on the objective from Lemma 2) in this setting. The absolute difference between outputs is smaller than $10^{-5}$ in all cases, verifying that minimum-cost flow solvers accurately measure smooth calibration.

Qualitatively, our results (based on smCE) agree with findings in [GPSW17] (based on binned variants of ECE), in that temperature scaling appears to be the most effective postprocessing technique.

**smCE tester.** Finally, we evaluated the efficiency of our proposed approaches to computing the empirical smCE. Specifically, we measure the runtime of four solvers for computing (3): a linear program solver from CVXPY, a commercial minimum-cost flow solver from Gurobi Optimization, a naïve implementation of our algorithm from Corollary 1 using Python, and a slightly-optimized implementation using the PyPy package [PyP19]. We use the same experimental setup as in Table 1, i.e. measuring calibration of a uniform predictor on a miscalibrated synthetic dataset, with $\varepsilon^\star = 0.01$.[12] In Table 3, we report the average runtimes for each trial (across 10 runs), varying the sample size. Again, the absolute difference between the outputs of all methods is negligible ($\leq 10^{-9}$ in all cases). As seen in Table 3, our custom algorithm (optimized with PyPy) outperforms standard packages from CVXPY and Gurobi Optimization starting from moderate sample sizes. We believe that Table 3 demonstrates that our new algorithms are a scalable, reliable way of testing calibration, and that these performance gains may be significantly improvable by further optimizing our code.

---

[12]We found similar runtime trends when using our algorithms to test calibration on the postprocessed neural network dataset, but the runtime gains were not as drastic as the sample size $n = 2^8$ was smaller in that case.

# Acknowledgements

We thank Edgar Dobriban for pointing us to the reference [LHHD23], and Yang P. Liu and Richard Peng for helpful discussions on the segment tree data structure in Section B.3. We also thank Yue Zhao for advice on running our experiments.

Table 3: Runtimes (in seconds) for computing the value of (3), using various solvers

| $n$ | $2^{10}$ | $2^{11}$ | $2^{12}$ | $2^{13}$ | $2^{14}$ | $2^{15}$ |
|---|---|---|---|---|---|---|
| CVXPY LP solver | 0.105 | 0.370 | 1.58 | 6.51 | 45.7 | 245 |
| Gurobi minimum-cost flow solver | 0.063 | 0.179 | 0.238 | 0.539 | 1.45 | 3.19 |
| Solver from Corollary 1 | 0.177 | 0.389 | 0.899 | 2.01 | 4.66 | 10.6 |
| Solver from Corollary 1 with PyPy | 0.079 | 0.115 | 0.176 | 0.307 | 0.621 | 2.05 |

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

# A Additional preliminaries

We first introduce some notation used in Appendices B, C, and D. When applied to a vector, we let $\|\cdot\|_p$ denote the $\ell_p$ norm for $p \geq 1$. We let $\mathbb{0}_d$ and $\mathbb{1}_d$ denote the all-zeroes and all-ones vectors in dimension $d$. We let $\Delta^d := \{x \in \mathbb{R}_{\geq 0}^d \mid \|x\|_1 = 1\}$ denote the probability simplex in dimension $d$. The $i^{\text{th}}$ coordinate basis vector is denoted $e_i$. We say $\tilde{x} \in \mathbb{R}$ is an $\varepsilon$-additive approximation of $x \in \mathbb{R}$ if $|\tilde{x} - x| \leq \varepsilon$. For a set $S \subset \mathbb{R}$, we say another set $T \subset \mathbb{R}$ is an $\varepsilon$-cover of $S$ if for all $s \in S$, there is $t \in T$ with $|s - t| \leq \varepsilon$. For $a, b \in \mathbb{R}$ with $a \leq b$, we let $\text{clip}_{[a,b]}(t) := \min(b, \max(a, t))$ denote the result of projecting $t \in \mathbb{R}$ onto $[a, b]$.

We next formally define our tolerant variant of the calibration testing problem in Definition 2.

**Definition 4** (Tolerant calibration testing). *Let $0 \leq \varepsilon_2 \leq \varepsilon_1 \leq 1$. We say algorithm $\mathcal{A}$ solves the $(\varepsilon_1, \varepsilon_2)$-tolerant calibration testing problem with $n$ samples, if given $n$ i.i.d. draws from a distribution $\mathcal{D}$ over $[0, 1] \times \{0, 1\}$, $\mathcal{A}$ returns either "yes" or "no" and satisfies the following with probability $\geq \frac{2}{3}$.*

- *$\mathcal{A}$ returns "no" if $\underline{\mathsf{dCE}}(\mathcal{D}) \geq \varepsilon_1$.*
- *$\mathcal{A}$ returns "yes" if $\underline{\mathsf{dCE}}(\mathcal{D}) \leq \varepsilon_2$.*

*In this case, we also call $\mathcal{A}$ an $(\varepsilon_1, \varepsilon_2)$-tolerant calibration tester.*

Note that an algorithm which solves the $(\varepsilon_1, \varepsilon_2)$-tolerant calibration testing problem with $n$ samples also solves the $\varepsilon_1$-calibration testing problem with the same sample complexity. Moreover, we give a simple impossibility result on parameter ranges for calibration testing.

**Lemma 5.** *Let $0 \leq \varepsilon_2 \leq \varepsilon_1 \leq \frac{1}{2}$ satisfy $\varepsilon_1 - \varepsilon_2 = \varepsilon$. There is a universal constant $C_{\text{coin}}$ such that, given $n \leq \frac{C_{\text{coin}}}{\varepsilon^2}$ samples from a distribution on $[0, 1] \times \{0, 1\}$, it is information-theoretically impossible to solve the $(\varepsilon_1, \varepsilon_2)$-tolerant calibration testing problem.*

*Proof.* Suppose $\varepsilon \leq \frac{1}{10}$, else we can choose $C_{\text{coin}}$ small enough such that $n < 1$. We consider two distributions over $[0, 1] \times \{0, 1\}$, $\mathcal{D}$ and $\mathcal{D}'$, with $\underline{\mathsf{dCE}}(\mathcal{D}) \geq \varepsilon_1$ but $\underline{\mathsf{dCE}}(\mathcal{D}') \leq \varepsilon_2$, so if $\mathcal{A}$ succeeds at tolerant calibration testing for both $\mathcal{D}$ and $\mathcal{D}'$, we must have $\mathrm{d}_{\mathrm{TV}}(\mathcal{D}^{\otimes n}, (\mathcal{D}')^{\otimes n}) \geq \frac{1}{3}$, where we denote the $n$-fold product of a distribution with $\cdot^{\otimes n}$. Else, $\mathcal{A}$ cannot return different answers from $n$ samples with probability $\geq \frac{2}{3}$, as required by Definition 4. Specifically, we define $\mathcal{D}, \mathcal{D}'$ as follows.

- To draw $(v, y) \sim \mathcal{D}$, let $v = \frac{1}{2} + \varepsilon_1$ and $y \sim \text{Bern}(\frac{1}{2})$.
- To draw $(v, y) \sim \mathcal{D}'$, let $v = \frac{1}{2} + \varepsilon_1$ and $y \sim \text{Bern}(\frac{1}{2} + \varepsilon)$.

We claim that $\underline{\mathsf{dCE}}(\mathcal{D}) = \varepsilon_1$. To see this, let $\Pi \in \text{ext}(\mathcal{D})$ and $(u, v, y) \sim \Pi$, so $\mathbb{E}_{(u,v,y) \sim \Pi}[u] = \frac{1}{2}$ because $u$ is calibrated. By Jensen's inequality, we have:

$$\mathbb{E}_{(u,v,y) \sim \Pi}[|u - v|] \geq \left|\mathbb{E}_{(u,v,y) \sim \Pi}[u] - \left(\frac{1}{2} + \varepsilon_1\right)\right| = \varepsilon_1.$$

The equality case is realized when $u = \frac{1}{2}$ with probability 1, proving the claim. Similarly, $\underline{\mathsf{dCE}}(\mathcal{D}') = \varepsilon_2$. Finally, let $\pi := \text{Bern}(\frac{1}{2})$, $\pi' := \text{Bern}(\frac{1}{2} + \varepsilon)$, and $\pi^{\otimes n}, (\pi')^{\otimes n}$ denote their $n$-fold product distributions. Pinsker's inequality shows that it suffices to show that $d_{\mathrm{KL}}(\pi^{\otimes n} \| (\pi')^{\otimes n}) \leq \frac{1}{5}$ to contradict our earlier claim $\mathrm{d}_{\mathrm{TV}}((\mathcal{D})^{\otimes n}, (\mathcal{D}')^{\otimes n}) \geq \frac{1}{3}$. To this end, we have

$$d_{\mathrm{KL}}(\pi^{\otimes n} \| (\pi')^{\otimes n}) = n \cdot d_{\mathrm{KL}}(\pi \| \pi') = \frac{n}{2}\left(\log\left(\frac{\frac{1}{2}}{\frac{1}{2} + \varepsilon}\right) + \log\left(\frac{\frac{1}{2}}{\frac{1}{2} - \varepsilon}\right)\right)$$

$$= \frac{n}{2}\log\left(\frac{1}{1 - 4\varepsilon^2}\right) \leq \frac{n}{2} \cdot 5\varepsilon^2 \leq \frac{1}{5},$$

where the first line used tensorization of $d_{\mathrm{KL}}$, and the last chose $C_{\text{coin}}$ small enough. $\square$

We also generalize Definitions 2 and 4 to apply to an arbitrary calibration measure.

**Definition 5** (d testing). *Let* d *be a calibration measure. For $\varepsilon \in \mathbb{R}_{\geq 0}$, we say algorithm $\mathcal{A}$ solves the $\varepsilon$-d-testing problem (or, $\mathcal{A}$ is an $\varepsilon$-d tester) with $n$ samples, if given $n$ i.i.d. draws from a distribution $\mathcal{D}$ over $[0,1] \times \{0,1\}$, $\mathcal{A}$ returns either "yes" or "no" and satisfies the following with probability $\geq \frac{2}{3}$.*

1. *$\mathcal{A}$ returns "no" if $\mathsf{d}(\mathcal{D}) \geq \varepsilon$.*

2. *$\mathcal{A}$ returns "yes" if $\mathsf{d}(\mathcal{D}) = 0$.*

*For $0 \leq \varepsilon_2 \leq \varepsilon_1$, we say algorithm $\mathcal{A}$ solves the $(\varepsilon_1, \varepsilon_2)$-tolerant d testing problem (or, $\mathcal{A}$ is an $(\varepsilon_1, \varepsilon_2)$-tolerant d tester) with $n$ samples, if given $n$ i.i.d. draws from a distribution $\mathcal{D}$ over $[0,1] \times \{0,1\}$, $\mathcal{A}$ returns either "yes" or "no" and satisfies the following with probability $\geq \frac{2}{3}$.*

1. *$\mathcal{A}$ returns "no" if $\mathsf{d}(\mathcal{D}) \geq \varepsilon_1$.*

2. *$\mathcal{A}$ returns "yes" if $\mathsf{d}(\mathcal{D}) \leq \varepsilon_2$.*

# B   Appendix for Section 2

## B.1   Deferred lemma proofs

*Proof of Lemma 1.* This follows from the triangle inequality:

$$|x_i - x_j| \leq \sum_{k=i}^{j-1} |x_k - x_{k+1}| \leq \sum_{k=i}^{j-1} v_{k+1} - v_k = v_j - v_i.$$

$\square$

## B.2   Dynamic programming

In this section, we give our dynamic programming approach to solving (4), which establishes Proposition 1. The graph $G$ in (4) is the union of a path $P$ on $n$ vertices and a star $S$ to the $(n+1)^{\text{th}}$ vertex. Let us identify the edges in $P$ with $[n-1]$ (where edge $i$ corresponds to vertices $(i, i+1)$), and the edges in $S$ with $[2n-1] \setminus [n-1]$. We first make a simplifying observation, which is that given the coordinates of a flow variable $f \in \mathbb{R}^E$ on the edges in the path $P$, there is a unique way to set the values of $f$ on the edges in the star $S$ so that the demands $\mathbf{B}^\top f = d$ are satisfied. Concretely, we require

$$f_{n-1+i} = d_i + f_i - f_{i-1} \text{ for all } 2 \leq i \leq n-1,$$
$$f_n = d_1 + f_1, \text{ and } f_{2n-1} = d_n - f_{n-1}.$$

Hence, minimizing the constrained problem in (4) is equivalent to minimizing the following unconstrained problem on the first $n-1$ flow variables, stated in (8) and reproduced here:

$$\min_{f \in \mathbb{R}^{n-1}} A(f) := |d_1 + f_1| + |d_n - f_{n-1}| + \sum_{i \in [n-2]} |f_i - f_{i+1} - d_{i+1}| + \sum_{i \in [n-1]} c_i |f_i|.$$

We now solve (8). We first define a sequence of partial functions $\{A_j : \mathbb{R} \to \mathbb{R}\}_{j \in [n-1]}$ by

$$A_1(z) := |d_1 + z| + c_1 |z|,$$
$$A_j(z) := \min_{\substack{f \in \mathbb{R}^j \\ f_j = z}} |d_1 + f_1| + \sum_{i \in [j-1]} |f_i - f_{i+1} - d_{i+1}| + \sum_{i \in [j-1]} c_i |f_i| \text{ for all } 2 \leq j \leq n-2,$$
$$A_{n-1}(z) := \min_{\substack{f \in \mathbb{R}^j \\ f_j = z}} A(f).$$

$$\tag{9}$$

In other words, $A_j(z)$ asks to minimize the partial function in (8) over the first $j$ flow variables $\{f_i\}_{i \in [j]}$, corresponding to all terms in which these flow variables participate, subject to fixing $f_j = z$. We make some preliminary observations about the partial functions $\{A_j\}_{j \in [n-1]}$.

**Lemma 6.** *For all $j \in [n-2]$, $A_j$ is a convex, continuous, piecewise linear function with at most $j+2$ pieces, and $A_{n-1}$ is a convex, continuous, piecewise linear function with at most $n+2$ pieces.*

*Proof.* We first establish convexity by induction; the base case $j = 1$ is clear. We next observe that

$$A_j(z) = c_j|z| + \min_{w \in \mathbb{R}} |w - z - d_j| + A_{j-1}(w) \text{ for all } 2 \leq j \leq n-2,$$

$$A_{n-1}(z) = |d_n - z| + c_{n-1}|z| + \min_{w \in \mathbb{R}} |w - z - d_{n-1}| + A_{n-2}(w). \tag{10}$$

In other words, each partial function $A_j$ can be recursively defined by first minimizing the first $j - 2$ flow variables for a fixed value $f_{j-1} = w$, and then taking the optimal choice of $w$. Moreover, supposing inductively $A_{j-1}$ is convex, $A_j$ is the sum of a convex function and a partial minimization over a jointly convex function of $(w, z)$, so it is also convex, completing the induction.

To see that $A_j$ is continuous (assuming continuity of $A_{j-1}$ inductively), it suffices to note $A_j$ is the sum of a continuous function and a partial minimization over a continuous function in two variables.

We now prove the claims about piecewise linearity, and the number of pieces. Clearly, $A_1$ is continuous and piecewise linear with at most 3 pieces. Next, for some $2 \leq j \leq n-2$, suppose $A_{j-1}$ is piecewise linear with vertices $\{v_i\}_{i \in [j]}$ in nondecreasing order (possibly with repetition) and slopes $\{t_i\}_{i=0}^{j}$, so $t_i$ is the slope of the segment of $A_{j-1}$ between $v_i$ and $v_{i+1}$, and $t_0$ and $t_j$ are the leftmost and rightmost slopes. For convenience we define $v_0 := -\infty$ and $v_{j+1} := \infty$. Consider the function

$$\min_{w \in \mathbb{R}} |w - z - d_j| + A_{j-1}(w). \tag{11}$$

For all values $z$ satisfying $v_i \leq z + d_j \leq v_{i+1}$ where $0 \leq i \leq j$, the function $|w - z - d_j| + A_{j-1}(w)$ is piecewise linear with vertices $v_1, \ldots, v_i, z + d_j, v_{i+1}, \ldots, v_j$, and correspondingly ordered slopes $t_0 - 1, t_1 - 1, \ldots, t_i - 1, t_i + 1, \ldots, t_j + 1$. The minimizing $w$ in (11) corresponds to any $v \in \{v_i\}_{i \in [j]} \cup \{z + d_j\}$ where the slope switches from nonpositive to nonnegative, which is either a fixed vertex $v = v_k$ for the entire range $v_i \leq z + d_j \leq v_{i+1}$, or the new vertex $z + d_j$ for this entire range.

In the former case, we have

$$\min_{w \in \mathbb{R}} |w - z - d_j| + A_{j-1}(w) = |v_k - z - d_j| + A_{j-1}(v_k), \tag{12}$$

which up to a constant additive shift is $|z - (v_k - d_j)|$, a linear function in $z$ in the range $v_i \leq z + d_j \leq v_{i+1}$ because the sign of $z - (v_k - d_j)$ does not change. In the latter case, we have

$$\min_{w \in \mathbb{R}} |w - z - d_j| + A_{j-1}(w) = A_{j-1}(z + d_j), \tag{13}$$

which again is a linear function in $z$ in the range $v_i \leq z + d_j \leq v_{i+1}$ by induction. In conclusion, (11) is linear in each range $z \in [v_i - d_j, v_{i+1} - d_j]$, so it is piecewise linear with at most $j + 1$ pieces; adding $c_j|z|$, which introduces at most 1 more piece, completes the induction. An analogous argument holds for $j = n - 1$, but we potentially introduce two more pieces due to adding $|d_n - z| + c_{n-1}|z|$.

For convenience, we now describe how to update the slopes and vertices going from the piecewise linear function $A_{j-1}$ to $A_j$. Also, as above suppose the vertices of $A_{j-1}$ are $\{v_i\}_{i \in [j]}$ and the corresponding slopes are $\{t_i\}_{i=0}^{j}$. Then because we argued (11) is linear in each range $z \in [v_i - d_j, v_{i+1} - d_j]$, it has vertices $\{v_i - d_j\}_{i \in [j]}$. If $z \in [v_i - d_j, v_{i+1} - d_j]$ and $t_i \in [-1, 1]$, then we are in the case of (13) and the corresponding slope in this range is $t_i$. Otherwise, if $t_i \leq -1$ we are in the case of (12) with slope $-1$, and if $t_i \geq 1$ the slope of the piece is similarly $1$. We then add $c_j|z|$ (and $|d_n - z|$ if $j = n - 1$). In summary, the new slopes and vertices are as follows.

- If $j \leq n - 2$, the new vertices are $\{v_i - d_j\}_{i \in [j]} \cup \{0\}$. If $v_k - d_j \leq 0 \leq v_{k+1} - d_j$ for some $0 \leq k \leq j$, using our convention $v_0 = -\infty$ and $v_{j+1} = \infty$, the new slopes are

$$\{\text{clip}_{[-1,1]}(t_i) - c_j\}_{0 \leq i < k} \cup \{\text{clip}_{[-1,1]}(t_k) - c_j\}$$
$$\cup \{\text{clip}_{[-1,1]}(t_k) + c_j\} \cup \{\text{clip}_{[-1,1]}(t_i) + c_j\}_{k < i \leq j}. \tag{14}$$

- If $j = n-1$, the new vertices are $\{v_i - d_j\}_{i \in [j]} \cup \{0\} \cup \{d_n\}$. If $v_k - d_j \le 0 \le v_{k+1} - d_j$ and $v_h - d_j \le d_n \le v_{h+1} - d_n$ for some $0 \le h, k \le j$, the new slopes are

$$\left\{\text{clip}_{[-1,1]}(t_i) - c_j \iota_{v_{i+1} \le d_j} + c_j \iota_{v_i \ge d_j}\right\}_{\substack{0 \le i \le j \\ i \notin \{h,k\}}}$$

$$\cup \left\{\text{clip}_{[-1,1]}(t_k) - c_j\right\} \cup \left\{\text{clip}_{[-1,1]}(t_k) + c_j\right\} \tag{15}$$

$$\cup \left\{\text{clip}_{[-1,1]}(t_h) - 1\right\} \cup \left\{\text{clip}_{[-1,1]}(t_h) + 1\right\},$$

where we let $\iota_{\mathcal{E}}$ denote the 0-1 indicator variable of an event $\mathcal{E}$.

The ordering of these vertices and their slopes are uniquely determined, because they are sorted similarly due to convexity, which implies nondecreasing slopes as $z$ increases. Finally, we note that assuming the invariant that at least one $\{t_i\}_{i=0}^j$ is nonnegative and at least one is nonpositive (which holds in the first iteration), in either of the cases (14) or (15) this invariant is preserved, since clipping preserves signs, the smallest slope decreases, and the largest slope increases. $\qquad\square$

We require one additional property of the slope updates (14).

**Lemma 7.** *For $j \in [n-1]$, let $\{t_i\}_{i=0}^j$ be the nondecreasing slopes of $A_j$. Then $t_0 \le -1$ and $t_j \ge 1$.*

*Proof.* By observation, the smallest and largest slopes of $A_1$ (9) are $-1 - c_1$ and $1 + c_1$. Hence, assuming inductively the lemma statement is true for $A_{j-1}$, the slope updates (14) result in smallest and largest slopes $-1 - c_j$ and $-1 + c_j$ in $A_j$, completing the induction. $\qquad\square$

We next observe that, by storing a constant amount of information in each iteration, we can work backwards from an optimal solution to $A_{n-1}$ and recover all flow variables which realized this value.

**Lemma 8.** *Let $\{v_i\}_{i \in [j]}$ and $\{t_i\}_{i=0}^j$ be the nondecreasing vertices and slopes of $A_{j-1}$ for some $2 \le j \le n-1$. Suppose we know $v_\ell$ and $v_r$ for $\ell, r \in [j]$, defined such that $t_{\ell-1} \le -1$ but $t_\ell \ge -1$, and similarly $t_{r-1} \le 1$ but $t_r \ge 1$. Then given a value of $z$, we can compute in $O(1)$ time*

$$\text{argmin}_{w \in \mathbb{R}} |w - z - d_j| + A_{j-1}(w).$$

*Proof.* Note that existence of $v_\ell, v_r$ is guaranteed by Lemma 7. We consider three cases.

First, if $z + d_j \in [v_\ell, v_r]$, we claim $w = z + d_j$. To see this, recall that if $z + d_j \in [v_i, v_{i+1}]$, we proved in Lemma 6 that the slopes of $|w - z - d_j| + A_{j-1}(w)$ (in $w$) are $t_0 - 1, t_1 - 1, \ldots, t_i - 1, t_i + 1, \ldots, t_j + 1$. Hence, the assumptions imply $t_i \in [-1, 1]$, so a vertex of $|w - z - d_j| + A_{j-1}(w)$ where the slope changes from nonpositive to nonnegative (i.e. the minimizing argument $w$) is $w = z + d_j$, as claimed.

To handle the other two cases, if $z + d_j \le v_\ell$, then the above calculation shows the new optimal vertex is $w = v_\ell$; similarly, if $z + d_j \ge v_r$ the new optimal vertex is $w = v_r$. $\qquad\square$

We now describe an interface for a data structure that we use to efficiently implement the updates (14) and (15), whose existence we prove in the following Appendix B.3.

**Lemma 9.** *There is a data structure, SegmentTree, which initializes a vector $t \in \mathbb{R}^n$ to $t \leftarrow \mathbb{0}_n$, and supports the following operations each in $O(\log n)$ time.*

- Query($i$): *Return $t_i$.*

- Add($\ell, r, c$): *Update $t_i \leftarrow t_i + c$ for all $\ell \le i \le r$.*

- Set($\ell, r, c$): *Update $t_i \leftarrow c$ for all $\ell \le i \le r$.*

Given access to SegmentTree, we now describe how to solve (8) in nearly-linear time, proving Proposition 1.

*Proof of Proposition 1.* We first describe how to compute all of the vertices and slopes $\{v_i\}_{i \in [n-1]}$, $\{t_i\}_{i=0}^{n-1}$ of $A_{n-2}$ in time $O(n \log^2(n))$. The proof of Lemma 6 shows that the vertices are

$$\left\{ -\sum_{k=j+1}^{n-2} d_k \right\}_{j=0}^{n-2}, \tag{16}$$

where we treat the empty sum as $0$. Moreover, the vertex $-\sum_{k=j+1}^{n-2} d_k$ is the new vertex which was inserted (as $0$) when computing $A_j$, and then advanced through the remaining iterations. We sort the vertices (16) into nondecreasing order, keeping track of the resulting permutation $\pi : [n-2] \cup \{0\} \to [n-1]$, i.e. if the vertex $-\sum_{k=j+1}^{n-2} d_k$ is the $i^{\text{th}}$ smallest in (16), then $\pi(j) = i$. This step takes time $O(n \log n)$ and does not dominate.

We next initialize a SegmentTree (Lemma 9) with $n$ vertices. We update it through the first $n-2$ recursive computations of slopes, via (14), keeping track of a time counter $j$, i.e. after we are done updating the slopes of $A_j$ the time counter increments. The state of SegmentTree at the end of time $j$ is as follows. For all $k \in [j]$, letting $\pi(h)$ be the next largest value after $\pi(k)$ amongst $\{\pi(k')\}_{k' \in [j]}$, we require that $t_i$ equals the slope of the segment to the right of the vertex inserted at time $k$ in $A_j$ for all the coordinates $\pi(k) + 1 \leq i \leq \pi(h)$. In other words, SegmentTree stores all of the slopes of $A_j$ in its coordinates (with redundancies due to vertices which will be inserted after time $j$), and $\pi$ maps the times vertices are inserted to their coordinate values in SegmentTree.

We next show how to maintain this state in $O(\log^2(n))$ time per time increment. In iteration $j$, the update (14) requires us to clip all previous slopes to the range $[-1, 1]$, subtract $c_j$ from all slopes in the range $[1, \pi(j)]$, and add $c_j$ to all slopes in the range $[\pi(j) + 1, n]$. We first use Query to perform binary searches for the values $\ell, r$ as defined in Lemma 7. We then sequentially apply

$$\mathsf{Set}(1, \ell, -1), \ \mathsf{Set}(r+1, n, 1), \ \mathsf{Add}(1, \pi(j), -c_j), \ \mathsf{Add}(\pi(j) + 1, n, c_j).$$

The dominant runtime term is the cost of $O(\log(n))$ calls to Query to perform the binary search, giving the claimed $O(\log^2(n))$ runtime per time increment. We can now call Query $n$ times to compute all slopes and vertices of $A_{n-2}$. We will also store the values of $v_\ell$ and $v_r$ at time $j$, which takes $O(1)$ time given $\ell, r, \pi$, and partial sums which can be precomputed in time $O(n)$.

Given these slopes and vertices, it is straightforward to apply (15) to compute all slopes and vertices of $A_{n-1}$ in time $O(n)$, at which point we can find $z := \text{argmin}_{z \in \mathbb{R}} A_{n-1}(z)$. Next, by using our stored values of $v_\ell$ and $v_r$ in every iteration, we can then use Lemma 8 to compute all the optimal flow values realizing $A_{n-1}(z)$. Finally, we can compute the optimal value (8) in time $O(n)$. $\qquad \square$

### B.3 Implementation of SegmentTree

In this section, we develop a data structure known as a *segment tree* which plays a vital role in our main algorithm. In particular, it allows us to prove Lemma 9. While this data structure is well-known folklore in the competitive programming community (see e.g. an overview of this technique in [QM22]), we provide a full description and proof for completeness.

For integers $\ell$ and $r$ satisfying $\ell \leq r$, we use $[\ell : r]$ to denote the set $\{\ell, \ell+1, \ldots, r\}$.

**Lemma 10** (Segment tree)**.** *Let $G$ be a semigroup with an identity element $e$, where the semigroup product of $a, b \in G$ is denoted by $a \cdot b$ or $ab$ and is not necessarily commutative. Let $v$ be an array of length $n$, where each element of $v$ is initialized to be the identity element $e$ of $G$. There is a data structure $\mathcal{D}$, called a* segment tree*, that can perform each of the following operations in $O(\log n)$ time (assuming a semigroup product can be computed in constant time).*

1. $\mathsf{Access}(i)$*: given $i \in [1 : n]$, return the $i^{th}$ element in $v$.*

2. $\mathsf{Apply}(g, \ell, r)$*: given $\ell, r \in [1 : n]$ satisfying $\ell \leq r$, and given a semigroup element $g \in G$, for each index $i \in [\ell : r]$, replace $v[i]$ with $g \cdot v[i]$.*

Before proving Lemma 10, we first use it to prove Lemma 9.

*Proof of Lemma 9.* We apply the data structure in Lemma 10 to a specific semigroup $G$ defined as follows. The elements of $G$ are functions $\tau : \mathbb{R} \to \mathbb{R}$, where the identity element $e$ is the identity

function $e(u) = u$ for every $u \in \mathbb{R}$, and the semigroup product is defined as function composition: $(a \cdot b)(u) = a(b(u))$ for every $a, b \in G$ and $u \in \mathbb{R}$. The semigroup $G$ consists of the following functions: $\mathsf{add}_c$ and $\mathsf{set}_c$ for every $c \in \mathbb{R}$. These functions are defined as follows:

$$\mathsf{add}_c(u) = u + c, \quad \mathsf{set}_c(u) = c, \quad \text{for every } u \in \mathbb{R}.$$

It is easy to check that these functions are closed under composition:

$$\mathsf{add}_c \cdot \mathsf{add}_{c'} = \mathsf{add}_{c+c'},$$
$$\mathsf{set}_c \cdot \mathsf{set}_{c'} = \mathsf{set}_c,$$
$$\mathsf{add}_c \cdot \mathsf{set}_{c'} = \mathsf{set}_{c+c'},$$
$$\mathsf{set}_c \cdot \mathsf{add}_{c'} = \mathsf{set}_c.$$

Therefore, $G$ is a valid semigroup. We can now implement the operations Query, Add, Set in Lemma 9 using the operations Access and Apply in Lemma 10 as follows.

To implement Query($i$), we run Access($i$) to obtain its output $g = v[i] \in G$, and return $g(0)$.

To implement Add($\ell, r, c$), we run Apply($\mathsf{add}_c, \ell, r$).

To implement Set($\ell, r, c$), we run Apply($\mathsf{set}_c, \ell, r$).

The correctness of this implementation can be shown inductively. At initialization, $v[i] = e$, and thus Query($i$) returns $e(0) = 0$, which is the correct value of $t_i$ at initialization. It remains to show inductively that after each Add and Set operation, the output of Query($i$), i.e., $v[i](0)$, is the intended value of $t_i$. Indeed, after an Add operation, for any $i \in [\ell : r]$, the element $v[i]$ is updated to $v[i]' := (\mathsf{add}_c \cdot v[i])$, and thus

$$v[i]'(0) = (\mathsf{add}_c \cdot v[i])(0) = \mathsf{add}_c(v[i](0)) = v[i](0) + c = t_i + c,$$

which is the intended new value of $t_i$. For $i \notin [\ell : r]$, the element $v[i]$ remains unchanged, and thus $v[i](0)$ remains unchanged. This is as desired because the new value of $t_i$ is intended to be the same as the old value. Combining these two cases, we have shown that the Add operation maintains that $v[i](0)$ is the intended value of $t_i$ for every $i \in [1 : n]$. We can similarly show that the Set operation also has this property, and thus our implementation is correct. The running time guarantee of the implementation follows directly from the running time guarantee in Lemma 10. $\qquad\square$

We prove Lemma 10 by describing the construction of the segment tree data structure and analyzing its correctness (Lemma 11) and efficiency (Lemma 12). By appending to the array an appropriate number of auxiliary entries, we can assume without loss of generality that the array length $n$ is a power of 2, i.e., $n = 2^r$ for a positive integer $r$. The data structure is implemented using a complete binary tree $T$ with depth $r$, where the $2^r$ leaves correspond to the $n = 2^r$ entries in the array.

More specifically, we use $\mathsf{seg}(\tau) \subseteq [1 : n]$ to denote the set of indices $i$ that a node $\tau \in T$ is associated with. For the root $\tau_0$ of the tree $T$, we set $\mathsf{seg}(\tau_0) = [1 : n] = [1 : 2^r]$, and for its two children $\tau_1, \tau_2$ we set $\mathsf{seg}(\tau_1) = [1 : n/2] = [1 : 2^{r-1}]$ and $\mathsf{seg}(\tau_2) = [n/2 + 1 : n] = [2^{r-1} + 1 : 2^r]$. In general, for any non-leaf node $\tau$ and its two children $\tau_1, \tau_2$, we set $\mathsf{seg}(\tau_1)$ as the first half of $\mathsf{seg}(\tau)$, and set $\mathsf{seg}(\tau_2)$ as the second half. In particular, for every leaf $\tau$, $\mathsf{seg}(\tau)$ is a singleton set consisting of a unique index $i \in [1 : n]$, and we say $\tau$ is the (unique) leaf corresponding to index $i$. See Figure 3 for an example with depth $r = 3$.

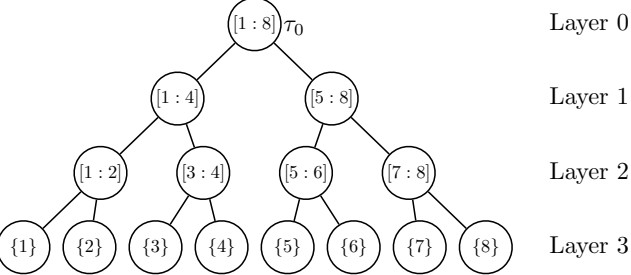

Figure 3: Example of segment tree with depth $r = 3$.

At every node $\tau$ of the tree, we maintain a semigroup element $g_\tau \in G$ initialized to be the identity element $e$.

**Access.**  We implement Access($i$) as follows. Let $\tau_0 \to \cdots \to \tau_r$ be the directed path from the root $\tau_0$ to the leaf $\tau_r$ corresponding to index $i$. Return the semigroup product $g_{\tau_0} g_{\tau_1} \cdots g_{\tau_r}$.

**Apply.**  We implement Apply($g, \ell, r$) recursively. That is, we implement Apply($g, \ell, r, \tau$) in Algorithm 1 which takes a node $\tau \in T$ as an additional input. We then define Apply($g, \ell, r$) to be Apply($g, \ell, r, \tau_0$), where the additional input is set as the root $\tau_0$ of the tree $T$.

---

**Algorithm 1** Apply($g, \ell, r, \tau$)

---

1: **if** $\mathrm{seg}(\tau) \cap [\ell : r] = \emptyset$ **then**
2:     **return**
3: **end if**
4: **if** $\mathrm{seg}(\tau) \subseteq [\ell : r]$ **then**
5:     $g_\tau \leftarrow g \cdot g_\tau$
6:     **return**
7: **end if**
8: Let $\tau_1, \tau_2$ be the two children of $\tau$
9: $g_{\tau_1} \leftarrow g_\tau \cdot g_{\tau_1}$
10: $g_{\tau_2} \leftarrow g_\tau \cdot g_{\tau_2}$
11: $g_\tau \leftarrow e$
12: Apply($g, \ell, r, \tau_1$)
13: Apply($g, \ell, r, \tau_2$)

---

**Correctness.**  At initialization, each array element $v[i]$ is initialized to be the identity element $e$. The semigroup element $g_\tau$ stored at each tree node $\tau$ is also initialized to be $e$, so Access($i$) returns the correct value $e$. It remains to show that Access($i$) still returns the correct value of $v[i]$ after each Apply operation. This is established in the following lemma:

**Lemma 11.** *For $i \in \{1, \ldots, n\}$, let $\tau_0 \to \cdots \to \tau_r$ be the directed path from the root $\tau_0$ to the leaf $\tau_r$ corresponding to index $i$. Let $\hat{g}_{\tau_0}, \ldots, \hat{g}_{\tau_r} \in G$ denote the current states of the semigroup elements $g_{\tau_0}, \ldots, g_{\tau_r}$ stored in the data structure. Then after we call Apply($g, l, r$), we have*

$$g_{\tau_0} \cdots g_{\tau_r} = \begin{cases} g \cdot \hat{g}_{\tau_0} \cdots \hat{g}_{\tau_r}, & \text{if } i \in [\ell : r]; \\ \hat{g}_{\tau_0} \cdots \hat{g}_{\tau_r}, & \text{if } i \notin [\ell : r]. \end{cases}$$

*Proof.* It is clear that $[1 : n] = \mathrm{seg}(\tau_0) \supseteq \cdots \supseteq \mathrm{seg}(\tau_r) = \{i\}$. If $i \in [\ell : r]$, let $\tau_j$ be the first node among $\tau_0, \ldots, \tau_r$ such that $\mathrm{seg}(\tau_j) \subseteq [\ell, r]$. We can inductively show that for every $j' = 1, \ldots, j$, right before we make the recursive call to Apply($g, \ell, r, \tau_{j'}$), we have

$$g_{\tau_i} = \begin{cases} e, & \text{if } i < j', \\ \hat{g}_{\tau_0} \cdots \hat{g}_{\tau_j}, & \text{if } i = j', \\ \hat{g}_{\tau_i}, & \text{if } i > j'. \end{cases}$$

When we call Apply($g, \ell, r, \tau_j$), since $\mathrm{seg}(\tau_j) \subseteq [\ell : r]$, Line 5 is executed and the function returns after that. Now we have

$$g_{\tau_i} = \begin{cases} e, & \text{if } i < j, \\ g \cdot \hat{g}_{\tau_0} \cdots \hat{g}_{\tau_j}, & \text{if } i = j, \\ \hat{g}_{\tau_i}, & \text{if } i > j. \end{cases}$$

This implies $g_{\tau_0} \cdots g_{\tau_r} = g \cdot \hat{g}_{\tau_0} \cdots \hat{g}_{\tau_r}$, as desired.

Similarly, if $i \notin [\ell, r]$, let $\tau_j$ be the first node among $\tau_0, \ldots, \tau_r$ such that $\mathrm{seg}(\tau_j) \cap [\ell : r] = \emptyset$. We can show that

$$g_{\tau_i} = \begin{cases} e, & \text{if } i < j; \\ \hat{g}_{\tau_0} \cdots \hat{g}_{\tau_j}, & \text{if } i = j; \\ \hat{g}_{\tau_i}, & \text{if } i > j. \end{cases}$$

This implies $g_{\tau_0} \cdots g_{\tau_r} = \hat{g}_{\tau_0} \cdots \hat{g}_{\tau_r}$, as desired. $\qquad\square$

**Efficiency.** The following result establishes the running time guarantee in Lemma 10.

**Lemma 12.** *Both Access and Apply run in $O(\log n)$ time.*

*Proof.* It is clear that Access runs in time $O(r) = O(\log n)$. When we run Apply$(g, \ell, r, \tau)$, the recursive calls at Lines 12-13 are made only when $[\ell : r]$ intersects but does not contain seg$(\tau)$, i.e., $\emptyset \subsetneq \text{seg}(\tau) \cap [\ell : r] \subsetneq \text{seg}(\tau)$. There are at most 2 such nodes $\tau$ at each level of the binary tree, so the total number of such nodes $\tau$ is $O(\log n)$. This implies that Apply runs in time $O(\log n)$. $\square$

### B.4 Tolerant testing via smooth calibration

For completeness, we first make the simple (but to our knowledge, new) observation that, while the constants in Lemma 4 are not necessarily tight, there is a constant gap between $\underline{\text{dCE}}$ and smCE.

**Lemma 13.** *Suppose for constants $B \geq A > 0$, it is the case that $A \cdot \underline{\text{dCE}}(\mathcal{D}) \leq \text{smCE}(\mathcal{D}) \leq B \cdot \underline{\text{dCE}}(\mathcal{D})$ for all distributions $\mathcal{D}$ over $[0, 1] \times \{0, 1\}$. Then, $\frac{B}{A} \geq \frac{3}{2}$.*

*Proof.* First, we claim that $A \leq 1$. To see this, let $(v, y) \sim \mathcal{D}$ be distributed where $v = \frac{1}{2}$ with probability 1, and $y \sim \text{Bern}(\frac{1}{2} + \varepsilon)$ for some $\varepsilon \in [0, \frac{1}{2}]$. Clearly, $\text{smCE}(\mathcal{D}) = |\frac{1}{2} - (\frac{1}{2} + \varepsilon)| = \varepsilon$. Moreover, $\underline{\text{dCE}}(\mathcal{D}) = \varepsilon$, which follows from the same Jensen's inequality argument as in Lemma 5, so this shows that $A \leq 1$. Next, we claim that $B \geq \frac{3}{2}$, concluding the proof. Consider the joint distribution over $(u, v, y)$ in Table B.4, and let $\mathcal{D}$ be the marginal of $(v, y)$. It is straightforward

| Probability mass | $u$ | $v$ | $y$ | $w(v)$ |
|---|---|---|---|---|
| $\frac{1}{2}$ | $\frac{1}{2}$ | $\frac{1}{2} - \varepsilon$ | 1 | 1 |
| $\frac{1}{2}$ | $\frac{1}{2}$ | $\frac{1}{2}$ | 0 | $1 - \varepsilon$ |

to check $(u, y)$ is calibrated and $\mathbb{E}|u - v| = \frac{\varepsilon}{2}$, so $\underline{\text{dCE}}(\mathcal{D}) \leq \frac{\varepsilon}{2}$. Moreover, $\text{smCE}(v, y) \geq \frac{3\varepsilon}{4}$, as witnessed by the Lipschitz weight function $w(v)$ in Table B.4, finishing our proof that $B \geq \frac{3}{2}$:

$$\text{smCE}(v, y) \geq \mathbb{E}[(y - v)w(v)] = \frac{1}{2}\left(\left(\frac{1}{2} + \varepsilon\right) \cdot 1\right) + \frac{1}{2}\left(\left(-\frac{1}{2}\right) \cdot (1 - \varepsilon)\right) = \frac{3\varepsilon}{4}.$$

$\square$

Using these claims, we now give our tolerant calibration tester in the regime $\varepsilon_1 > 4\varepsilon_2$.

**Theorem 3.** *Let $0 \leq \varepsilon_2 \leq \varepsilon_1 \leq 1$ satisfy $\varepsilon_1 > 4\varepsilon_2$, and let $n \geq C_{\text{tct}} \cdot \frac{1}{(\varepsilon_1 - 4\varepsilon_2)^2}$ for a universal constant $C_{\text{tct}}$. There is an algorithm $\mathcal{A}$ which solves the $(\varepsilon_1, \varepsilon_2)$-tolerant calibration testing problem with $n$ samples, which runs in time*

$$O\left(n \log^2(n)\right).$$

*Proof.* Throughout the proof, let $\alpha := \frac{\varepsilon_1}{2} - 2\varepsilon_2 > 0$. Consider the following algorithm.

1. Sample $n \geq \frac{C_{\text{tct}}}{\alpha^2}$ samples to form an empirical distribution $\widehat{\mathcal{D}}_n$, where $C_{\text{tct}}$ is chosen large enough so that Lemma 3 guarantees $|\text{smCE}(\mathcal{D}) - \text{smCE}(\widehat{\mathcal{D}}_n)| \leq \frac{\alpha}{2}$ with probability $\geq \frac{2}{3}$.

2. Call Corollary 1 to obtain $\beta$, the value of $\text{smCE}(\widehat{\mathcal{D}}_n)$.

3. Return "yes" if $\beta \leq 2\varepsilon_2 + \frac{\alpha}{2}$, and return "no" otherwise.

Conditioned on the event $|\text{smCE}(\mathcal{D}) - \text{smCE}(\widehat{\mathcal{D}}_n)| \leq \frac{\alpha}{2}$, we show that the algorithm succeeds in tolerant calibration testing. First, if $\underline{\text{dCE}}(\mathcal{D}) \leq \varepsilon_2$, then $\text{smCE}(\mathcal{D}) \leq 2\varepsilon_2$ by Lemma 4, and therefore by the assumed success of Lemma 3, the algorithm will return "yes." Second, if $\underline{\text{dCE}}(\mathcal{D}) \geq \varepsilon_1$, then $\text{smCE}(\mathcal{D}) \geq \frac{\varepsilon_1}{2}$ by Lemma 4, and similarly the algorithm returns "no" in this case. Finally, the runtime is immediate from Corollary 1; all other steps take $O(1)$ time. $\square$

# C Lower distance to calibration

In this section, we provide our main result on approximating the lower distance to calibration of a distribution on $[0, 1] \times \{0, 1\}$. We provide details on a framework for lifting constrained linear programs to equivalent unconstrained counterparts in Appendix C.1. In Appendix C.2, we next state preliminary definitions and results from [BGHN23a] used in our algorithm. In Appendix C.3, we then develop a rounding procedure compatible with a linear program which closely approximates the empirical lower distance to calibration. Finally, in Appendix C.4, we use our rounding procedure to design an algorithm for calibration testing, which solves the problem for a larger range of parameters than Theorem 4 (i.e. the entire relevant parameter range), at a quadratic runtime overhead.

## C.1 Rounding linear programs

In this section, we give a general framework for approximately solving linear programs, following similar developments in the recent combinatorial optimization literature [She13, JST19]. Roughly speaking, this framework is a technique for losslessly converting a constrained convex program to an unconstrained one, provided we can show existence of a rounding procedure compatible with the constrained program in an appropriate sense. We begin with our definition of a rounding procedure.

**Definition 6** (Rounding procedure). *Consider a convex program defined on the intersection of convex set $\mathcal{X}$ with linear equality constraints $\mathbf{A}x = b$:*

$$\min_{\substack{x \in \mathcal{X} \\ \mathbf{A}x = b}} c^\top x. \tag{17}$$

*We say* Round *is a $(\widetilde{\mathbf{A}}, \tilde{b}, p)$-equality rounding procedure for $(\mathbf{A}, b, c, \mathcal{X})$ if $p \geq 1$, and for any $x \in \mathcal{X}$, there exists $x' := \mathsf{Round}(x) \in \mathcal{X}$ such that $\mathbf{A}x' = b$, $\widetilde{\mathbf{A}}x' = \tilde{b}$, and*

$$c^\top x' \leq c^\top x + \left\| \widetilde{\mathbf{A}}x - \tilde{b} \right\|_p. \tag{18}$$

Intuitively, rounding procedures replace the hard-constrained problem (17) with its soft-constrained variants, i.e. the soft equality-constrained

$$\min_{x \in \mathcal{X}} c^\top x + \left\| \widetilde{\mathbf{A}}x - \tilde{b} \right\|_p, \tag{19}$$

for some $(\widetilde{\mathbf{A}}, \tilde{b}, p)$ constructed from the corresponding hard-constrained problem instance (parameterized by $\mathbf{A}, b, c, \mathcal{X}$). Leveraging the assumptions on our rounding procedure, we now show how to relate approximate solutions to these problems, generalizing Lemma 1 of [JST19].

**Lemma 14.** *Let $x$ be an $\varepsilon$-approximate minimizer to (19), and let* Round *be a $(\widetilde{\mathbf{A}}, \tilde{b}, p)$-equality rounding procedure for $(\mathbf{A}, b, c, \mathcal{X})$. Then $x' := \mathsf{Round}(x)$ is an $\varepsilon$-approximate minimizer to (17).*

*Proof.* We first claim that a minimizing solution to (19) satisfies the constraints $\mathbf{A}x = b$. To see this, given any $x \in \mathcal{X}$, we can produce $x' \in \mathcal{X}$ with $\mathbf{A}x' = b$ and such that $x'$ has smaller objective value in (19). Indeed, letting $x' := \mathsf{Round}(x')$, (18) guarantees

$$c^\top x' = c^\top x' + \left\| \widetilde{\mathbf{A}}x' - \tilde{b} \right\|_p \leq c^\top x + \left\| \widetilde{\mathbf{A}}x - \tilde{b} \right\|_p,$$

as claimed. Now let $x^\star \in \mathcal{X}$ satisfying $\mathbf{A}x^\star = b$ minimize (19). Then, if $x$ is an $\varepsilon$-approximate minimizer to (19) and $x' = \mathsf{Round}(x)$, we have the desired claim from $\mathbf{A}x' = b$, and

$$c^\top x' = c^\top x' + \left\| \widetilde{\mathbf{A}}x' - \tilde{b} \right\|_p \leq c^\top x + \left\| \widetilde{\mathbf{A}}x - \tilde{b} \right\|_p \leq c^\top x^\star + \left\| \widetilde{\mathbf{A}}x^\star - \tilde{b} \right\|_p = c^\top x^\star + \varepsilon.$$

$\square$

In the remainder of the section, we apply our rounding framework to a hard-constrained linear program, in the $p = 1$ geometry. To aid in approximately solving the soft-constrained linear programs arising from our framework, we use the following procedure from [JT23], building upon the recent

literature for solving box-simplex games at accelerated rates [She17, JST19, CST21]. In the statement of Proposition 2, we use the notation

$$\|\mathbf{A}\|_{p \to q} := \max_{x \in \mathbb{R}^n |\|x\|_p \leq 1} \|\mathbf{A}x\|_q.$$

Notice that in particular, $\|\mathbf{A}\|_{1 \to 1}$ is the largest $\ell_1$ norm of any column of $\mathbf{A}$.

**Proposition 2** (Theorem 1, [JT23]). *Let* $\mathbf{A} \in \mathbb{R}^{n \times d}$, $b \in \mathbb{R}^d$, $c \in \mathbb{R}^n$, *and* $\varepsilon > 0$. *There is an algorithm which computes an* $\varepsilon$-*approximate saddle point to the* box-simplex game

$$\min_{x \in [-1,1]^n} \max_{y \in \Delta^d} x^\top \mathbf{A}y - b^\top y + c^\top x, \tag{20}$$

*in time*

$$O\left(\text{nnz}(\mathbf{A}) \cdot \frac{\|\mathbf{A}\|_{1 \to 1} \log d}{\varepsilon}\right).$$

We also require a standard claim on converting minimax optimization error to error on an induced minimization objective. To introduce our notation, we say that $x \in \mathcal{X}$ is an $\varepsilon$-approximate minimizer of $f : \mathcal{X} \to \mathbb{R}$ if $f(x) - \min_{x' \in \mathcal{X}} f(x') \leq \varepsilon$. We call $(x, y) \in \mathcal{X} \times \mathcal{Y}$ an $\varepsilon$-approximate saddle point to a convex-concave function $f : \mathcal{X} \times \mathcal{Y} \to \mathbb{R}$ if its duality gap is at most $\varepsilon$, i.e.

$$\max_{y' \in \mathcal{Y}} f(x, y') - \min_{x' \in \mathcal{X}} f(x', y) \leq \varepsilon.$$

**Lemma 15.** *Let* $f : \mathcal{X} \times \mathcal{Y} \to \mathbb{R}$ *be convex-concave for compact* $\mathcal{X}, \mathcal{Y}$, *and let* $g(x) := \max_{y \in \mathcal{Y}} f(x, y)$ *for* $x \in \mathcal{X}$. *If* $(x, y)$ *is an* $\varepsilon$-*approximate saddle point to* $f$, $x$ *is an* $\varepsilon$-*approximate minimizer to* $g$.

*Proof.* Let $y' := \text{argmax}_{y \in \mathcal{Y}} f(x, y)$ and $x' := \text{argmin}_{x' \in \mathcal{X}} f(x', y)$. The conclusion follows from

$$\min_{x^\star \in \mathcal{X}} g(x^\star) = \min_{x^\star \in \mathcal{X}} \max_{y^\star \in \mathcal{Y}} f(x^\star, y^\star) = \max_{y^\star \in \mathcal{Y}} \min_{x^\star \in \mathcal{X}} f(x^\star, y^\star) \geq \min_{x' \in \mathcal{X}} f(x', y),$$

where we used strong duality (via Sion's minimax theorem), so that

$$g(x) - \min_{x^\star \in \mathcal{X}} g(x^\star) \leq g(x) - f(x', y) = f(x, y') - f(x', y) \leq \varepsilon.$$

$\square$

The following corollary of Proposition 2 will be particularly useful in our development, which is immediate using Lemma 15, upon negating the box-simplex game (20), exchanging the names of the variables $(x, y)$, $(b, c)$, and explicitly maximizing over $y \in [-1, 1]^n$, i.e.

$$\min_{x \in \Delta^d} \langle c, x \rangle + \|\mathbf{A}x - b\|_1 = \min_{x \in \Delta^d} \max_{y \in [-1,1]^n} \langle c, x \rangle + y^\top (\mathbf{A}x - b).$$

**Corollary 2.** *Let* $\mathbf{A} \in \mathbb{R}^{n \times d}$, $b \in \mathbb{R}^n$, $c \in \mathbb{R}^d$, *and* $\varepsilon > 0$. *There is an algorithm which computes an* $\varepsilon$-*approximate minimizer to* $\min_{x \in \Delta^d} c^\top x + \|\mathbf{A}x - b\|_1$, *in time*

$$O\left(\text{nnz}(\mathbf{A}) \cdot \frac{\|\mathbf{A}\|_{1 \to 1} \log d}{\varepsilon}\right).$$

### C.2 LDTC preliminaries

In this section, we collect preliminaries for our testing algorithm based on estimating the LDTC. First, analogously to Lemma 3, we recall a bound from [BGHN23a] on the deviation of the empirical estimate of $\underline{\text{dCE}}(\mathcal{D})$ from the population truth which holds with constant probability.

**Lemma 16** (Theorem 9.10, [BGHN23a]). *For any* $\varepsilon \in (0, 1)$, *there is an* $n = O(\frac{1}{\varepsilon^2})$ *such that, if* $\widehat{\mathcal{D}}_n$ *is the empirical distribution over* $n$ *i.i.d. draws from* $\mathcal{D}$, *with probability* $\geq \frac{2}{3}$,

$$\left|\underline{\text{dCE}}(\mathcal{D}) - \underline{\text{dCE}}(\widehat{\mathcal{D}}_n)\right| \leq \varepsilon.$$

Next, given a set $U \subset [0, 1]$, we provide an analog of Definition 1 which is restricted to $U$.

**Definition 7** (*U*-LDTC)**.** *Let $U \subset [0,1]$, and let $\mathcal{D}$ be a distribution over $[0,1] \times \{0,1\}$. Define $\text{ext}^U(\mathcal{D})$ to be all joint distributions $\Pi$ over $(u,v,y) \in U \times [0,1] \times \{0,1\}$, with the following properties.*

- *The marginal distribution of $(v,y)$ is $\mathcal{D}$.*

- *The marginal distribution $(u,y)$ is perfectly calibrated, i.e. $\mathbb{E}_\Pi[y|u] = u$.*

*The $U$-lower distance to calibration (U-LDTC) of $\mathcal{D}$, denoted $\underline{\mathsf{dCE}}^U(\mathcal{D})$, is defined by*

$$\underline{\mathsf{dCE}}^U(\mathcal{D}) := \inf_{\Pi \in \text{ext}^U(\mathcal{D})} \mathbb{E}_{(u,v,y)\sim\Pi} |u - v|.$$

Note that if we require $\{0,1\} \subset U$, then $\text{ext}^U(\mathcal{D})$ is always nonempty, because we can let $u = y$ with probability 1. We also state a helper claim from [BGHN23a], which relates $\underline{\mathsf{dCE}}^U$ to $\underline{\mathsf{dCE}}$.

**Lemma 17** (Lemma 7.11, [BGHN23a])**.** *Let $\mathcal{D}$ be a distribution over $[0,1] \times \{0,1\}$, and let $U$ be a finite $\frac{\varepsilon}{2}$-covering of $[0,1]$ satisfying $\{0,1\} \subseteq U$. Then, $\underline{\mathsf{dCE}}(\mathcal{D}) \leq \underline{\mathsf{dCE}}^U(\mathcal{D}) \leq \underline{\mathsf{dCE}}(\mathcal{D}) + \varepsilon.$*

To this end, in the rest of the section we define, for any $\varepsilon \in (0,1)$,

$$U_\varepsilon := \{0,1\} \cup \left\{ \frac{i\varepsilon}{2} \mid i \in \left[ \left\lfloor \frac{2}{\varepsilon} \right\rfloor \right] \right\}, \tag{21}$$

which is an $\frac{\varepsilon}{2}$-cover of $[0,1]$ satisfying $|U_\varepsilon| = O(\frac{1}{\varepsilon})$. Finally, we state a linear program, derived in [BGHN23a], whose value equals $\underline{\mathsf{dCE}}^U(\mathcal{D})$, when the first marginal of $\mathcal{D}$ is discretely supported.

**Lemma 18** (Lemma 7.6, [BGHN23a])**.** *Let $U, V \subset [0,1]$ be discrete sets, where $\{0,1\} \subset U$, and let $\mathcal{D}$ be a distribution over $V \times \{0,1\}$, where for $(v,y) \in V \times \{0,1\}$ we denote the probability of $(v,y) \sim \mathcal{D}$ by $\mathcal{D}(v,y)$. The following linear program with $2|U||V|$ variables $\Pi(u,v,y)$ for all $(u,v,y) \in U \times V \times \{0,1\}$, is feasible, and its optimal value equals $\underline{\mathsf{dCE}}^U(\mathcal{D})$:*

$$\min_{\Pi \in \mathbb{R}_{\geq 0}^{2|U||V|}} \sum_{(u,v,y)\in U \times V \times \{0,1\}} |u - v| \, \Pi(u,v,y)$$

*such that* $\sum_{u \in U} \Pi(u,v,y) = \mathcal{D}(v,y)$, *for all* $(v,y) \in V \times \{0,1\}$,

*and* $(1 - u) \sum_{v \in V} \Pi(u,v,1) = u \sum_{v \in V} \Pi(u,v,0)$, *for all* $u \in U$.

### C.3 Rounding for empirical $U$-LDTC

In this section we fix a dataset under consideration,

$$\widehat{\mathcal{D}}_n := \{(v_i, y_i)\}_{i \in [n]} \subset [0,1] \times \{0,1\},$$

and the corresponding empirical distribution, also denoted $\widehat{\mathcal{D}}_n$, where $(v,y) \sim \widehat{\mathcal{D}}_n$ means $(v,y) = (v_i, y_i)$ with probability $\frac{1}{n}$ for each $i \in [n]$. We let $V := \{v_i\}_{i \in [n]}$ be identified with $[n]$ in the natural way. Moreover, for a fixed parameter $\varepsilon \in (0,1)$ throughout, we let $U := U_\varepsilon$ defined in (21). Finally, we denote $m := |U| = O(\frac{1}{\varepsilon})$, and let $\mathbf{U} \in [0,1]^{m \times m}$ be the diagonal matrix whose diagonal entries correspond to $U$. We also identify elements of $U$ with $j \in [m]$ in an arbitrary but consistent way, writing $u_j \in [0,1]$ to mean the $j^{\text{th}}$ element of $U$ according to this identification.

We next rewrite the linear program in Lemma 18 into a more convenient reformulation.

**Lemma 19.** *The linear program in Lemma 18 can equivalently be written as:*

$$\underline{\mathsf{dCE}}^U(\widehat{\mathcal{D}}_n) := \min_{\substack{x \in \mathcal{X} \\ \mathbf{M}x = \frac{1}{n}\mathbb{1}_n \\ \mathbf{U}\mathbf{B}_0 x_0 = (\mathbf{I}_m - \mathbf{U})\mathbf{B}_1 x_1}} c^\top x, \quad \text{where } \mathcal{X} := \Delta^{2mn} \text{ and we denote } x = \begin{pmatrix} x_0 \in \mathbb{R}^{mn} \\ x_1 \in \mathbb{R}^{mn} \end{pmatrix},$$

$$\tag{22}$$

*where we define $c \in \mathbb{R}^{2mn}$, $\mathbf{M} \in \mathbb{R}^{n \times 2mn}$, and $\mathbf{B}_0, \mathbf{B}_1 \in \mathbb{R}^{m \times mn}$ by*

$$c_{(i,j,k)} := |u_j - v_i| \quad \textit{for all } (i,j,k) \in [n] \times [m] \times \{0,1\},$$

$$\mathbf{M}_{i',(i,j,k)} := \begin{cases} 1 & y_i = k, \ i' = i \\ 0 & else \end{cases} \quad \textit{for all } i' \in [n], (i,j,k) \in [n] \times [m] \times \{0,1\},$$

$$\textit{and } [\mathbf{B}_0]_{j',(i,j,k)} := \begin{cases} 1 & j = j', \ k = 0 \\ 0 & else \end{cases} \quad \textit{for all } j' \in [m], (i,j,k) \in [n] \times [m] \times \{0,1\},$$

$$[\mathbf{B}_1]_{j',(i,j,k)} := \begin{cases} 1 & j = j', \ k = 1 \\ 0 & else \end{cases} \quad \textit{for all } j' \in [m], (i,j,k) \in [n] \times [m] \times \{0,1\}.$$

*Proof.* This is clear from observation, but we give a brief explanation of the notation. First, $x \in \mathcal{X}$ represents the density function of our joint distribution $\Pi$ over $U \times V \times \{0,1\}$, and has $2mn$ coordinates identified with elements $(i,j,k) \in V \times U \times \{0,1\} \equiv [n] \times [m] \times \{0,1\}$. We let the subset of coordinates with $k = 0$ be denoted $x_0 \in \mathbb{R}^{mn}$, defining $x_1$ similarly. Recalling the definition of the linear program in Lemma 18, $x_{(i,j,k)}$ is indeed reweighted by $c_{(i,j,k)} = |u_j - v_i|$.

Next, $\mathbf{M}$ represents the marginal constraints in Lemma 18, and enforcing $\mathbf{M}x = \frac{1}{n}\mathbb{1}_n$ is equivalent to the statement that, for each $i' \in [n]$, the sum of all entries $(i,j,k)$ of $x$ with $i = i'$ and $k = y_i$ is $\frac{1}{n}$, since that is the probability density assigned to $(v_{i'}, y_{i'})$ by the distribution $\widehat{\mathcal{D}}_n$.

Lastly, the $j^{\text{th}}$ calibration constraint in Lemma 18 is enforced by the $j^{\text{th}}$ row of the equation $\mathbf{U}\mathbf{B}_0 x_0 = (\mathbf{I}_m - \mathbf{U})\mathbf{B}_1 x_1$, which reads $u_j \langle [\mathbf{B}_0]_{j:}, x_0 \rangle = (1 - u_j) \langle [\mathbf{B}_1]_{j:}, x_1 \rangle$. We can check by the definitions of $[\mathbf{B}_0]_{j:}, [\mathbf{B}_1]_{j:}$ that this is consistent with our earlier calibration constraints.

$\square$

We give a convenient way of visualizing the marginal and calibration constraints described in Lemma 19. For convenience, we identify each $x \in \Delta^{2mn}$ with an $m \times 2n$ matrix

$$\text{mat}(x) = \mathbf{X} = \begin{pmatrix} \mathbf{X}_0 \in \mathbb{R}^{m \times n} & \mathbf{X}_1 \in \mathbb{R}^{m \times n} \end{pmatrix}, \tag{23}$$

where $\mathbf{X}_0$ consists of entries of $x_0$ arranged in a matrix fashion (with rows corresponding to $[m] \equiv U$ and columns corresponding to $[n] \equiv V$), and similarly $\mathbf{X}_1$ is a rearrangement of $x_1$, recalling (22). When explaining how we design our rounding procedures to modify $\mathbf{X}$ to satisfy constraints, it will be helpful to view entries of $\mathbf{X}$ as denoting an amount of physical mass which we can move around.

There are $2n$ columns in $\mathbf{X}$, corresponding to pairs $(i,k) \in V \times \{0,1\}$; among these, we say $n$ columns are "active," where column $(i,k)$ is active iff $y_i = k$, and we say the other $n$ columns are "inactive." Following notation (23), the marginal constraints $\mathbf{X} = \frac{1}{n}\mathbb{1}_n$ simply ask that the total amount of mass in each active column is $\frac{1}{n}$, so there is no mass in any inactive column since $x \in \Delta^{2mn}$.

Moreover, there are $m$ rows in $\mathbf{X}$, each corresponding to some $j \in U$. If we let $\ell_j$ denote the amount of mass on $[\mathbf{X}_0]_{j:}$ and $r_j$ the amount of mass on $[\mathbf{X}_1]_{j:}$, the $j^{\text{th}}$ calibration constraint simply asks that $u_j \ell_j = (1 - u_j)r_j$, i.e. it enforces balance on the amount of mass in each row's two halves.

Finally, for consistency with Definition 6, the linear program in (22) can be concisely written as

$$\min_{\substack{x \in \mathcal{X} \\ \mathbf{A}x = b}} c^\top x, \text{ where } \mathbf{A} := \begin{pmatrix} \mathbf{M} \\ \mathbf{B} \end{pmatrix}, \ \mathbf{B} := \begin{pmatrix} \mathbf{U}\mathbf{B}_0 & -(\mathbf{I}_m - \mathbf{U})\mathbf{B}_1 \end{pmatrix}, \ b := \begin{pmatrix} \frac{1}{n}\mathbb{1}_n \\ \mathbb{0}_m \end{pmatrix} \tag{24}$$

and $c, \mathcal{X}$ are as defined in (22). In the rest of the section, following Definition 6, we develop an equality rounding procedure for the equality-constrained linear program in (24) in two steps.

1. In Lemma 20, we first show how to take $x \in \mathcal{X}$ with $\|\mathbf{M}x - \frac{1}{n}\mathbb{1}_n\|_1 = \Delta$, and produce $x' \in \mathcal{X}$ such that $\mathbf{M}x' = \frac{1}{n}\mathbb{1}_n$ (i.e. $x'$ now satisfies the marginal constraints) and $\|x - x'\|_1 = O(\Delta)$.

2. In Lemma 22, we then consider $x \in \mathcal{X}$ such that, following the notation (22), $\|\mathbf{U}\mathbf{B}_0 x_0 - (\mathbf{I}_m - \mathbf{U})\mathbf{B}_1 x_1\|_1 = \Delta$. We show how to produce $x' \in \mathcal{X}$ such that $\mathbf{M}x = \mathbf{M}x'$ (i.e. the marginals of $x'$ are unchanged), $\mathbf{U}\mathbf{B}_0 x'_0 = (\mathbf{I}_m - \mathbf{U})\mathbf{B}_1 x'_1$ (i.e. $x'$ is calibrated), and $\langle c, x' - x \rangle = O(\Delta)$.

Our rounding procedure uses Lemma 20 to satisfy the marginal constraints in (22), and then applies Lemma 22 to the result to satisfy the calibration constraints in (22) without affecting the marginal constraints. By leveraging the stability guarantees on these steps, we can show this is indeed a valid rounding procedure in the sense of (18). We now give our first step for marginal satisfaction.

**Lemma 20** (Marginal satisfaction). *Following notation in* (22), *let $x \in \mathcal{X}$ satisfy $\|\mathbf{M}x - \frac{1}{n}\mathbb{1}_n\|_1 = \Delta$. There is an algorithm which runs in time $O(mn)$, and returns $x'$ with*

$$\mathbf{M}x' = \frac{1}{n}\mathbb{1}_n, \ \|x - x'\|_1 \le 2\Delta.$$

*Proof.* Recall for $i \in [n]$, we say column $i$ of $\mathbf{X}_0$ is active if $y_i = 0$, and similarly column $i$ of $\mathbf{X}_1$ is active if $y_i = 1$. We call $I$ the set of $n$ inactive columns, and partition $A$, which we call the set of $n$ active columns, into three sets $A^>$, $A^=$, and $A^<$, where $A^>$ are the columns whose sums are $> \frac{1}{n}$, $A^<$ are the columns whose sums are $< \frac{1}{n}$, and $A^=$ are the remaining columns. Hence, every column of $\mathbf{X}$ belongs to $I$, $A^>$, $A^=$, or $A^\le$. Note that until $|A^=| = n$, we can never have $A^< = \emptyset$, since this means all column sums in $A$ are $\ge \frac{1}{n}$ (with at least 1 strict inequality), contradicting $x \in \mathcal{X}$.

We first take columns $i \in A^>$ one at a time, and pair them with an arbitrary column in $i' \in A^<$, moving mass from column $i$ arbitrarily to column $i'$ until either column $i$ or column $i'$ enters $A^=$. We charge this movement to the marginal constraints corresponding to $i$ and $i'$, since the constraints were violated by the same amount as the mass being moved. After this process is complete, $A^>$ is empty, and we only moved mass from columns originally in $A^>$ to columns originally in $A^<$.

Next, we take columns $i \in I$ one at a time, and pair them with an arbitrary column $i' \in A^<$, moving mass until either column $i$ is $\mathbb{0}_m$ or column $i'$ enters $A^=$. We can charge half this movement to the marginal constraint corresponding to $i'$, since the sign of the marginal violation stays the same throughout. Hence, the overall movement is $\le 2\Delta$. After this is complete, all columns in $I$ are $\mathbb{0}_m$ and all columns in $A$ are in $A^=$, so we can return $x' \in \Delta^{2mn}$ corresponding to the new matrix.

It is clear both steps of this marginal satisfaction procedure take $O(mn)$ time, since we can sequentially process columns in $A^-$ until they enter $A^=$, and will never be considered again. $\square$

We next describe a procedure which takes $x \in \Delta^{2mn}$, and modifies it to satisfy the calibration constraints $\mathbf{U}\mathbf{B}_0 x = (\mathbf{I}_m - \mathbf{U})\mathbf{B}_1 x$ without changing the marginals $\mathbf{M}x$. We first provide a helper lemma used in our rounding procedure, which describes how to fix the $j^{\text{th}}$ marginal constraint.

**Lemma 21.** *Let $x \in \Delta^{2mn}$ and $\mathbf{X} := \text{mat}(x)$ as defined in* (23). *Let $j \in [m]$ correspond to an element $u_j \in U$, let $\ell_j := \|[\mathbf{X}_0]_{j:}\|_1$, $r_j := \|[\mathbf{X}_1]_{j:}\|_1$, and let $\Delta_j := |u_j\ell_j - (1-u_j)r_j|$. There exists $j' \in [m]$ such that we can move mass from only $\mathbf{X}_{j:}$ to $\mathbf{X}_{j':}$, resulting in $\mathbb{R}^{m \times 2n} \ni \mathbf{X}' \equiv x' \in \Delta^{2mn}$ such that $\mathbf{M}x' = \mathbf{M}x$, $u_j\|[\mathbf{X}'_0]_{j:}\|_1 = (1-u_j)\|[\mathbf{X}'_1]_{j:}\|_1$, and $\langle c, x' - x \rangle \le \Delta_j$.*

*Proof.* Without loss of generality, suppose that the row $j = 1$ corresponds to $u_j = 0$, and $j = m$ corresponds to $u_j = 1$. We split the proof into two cases, depending on the sign of $u_j\ell_j - (1-u_j)r_j$.

*Case 1: $u_j\ell_j > (1-u_j)r_j$.* We let $j' = 1$, i.e. we only move mass from the $j^{\text{th}}$ row to the first row. Specifically, we leave $[\mathbf{X}_1]_{j:}$ unchanged, and move mass from $[\mathbf{X}_0]_{j:}$ to $[\mathbf{X}_0]_{1:}$, making sure to only move mass in the same column. The total amount of mass we must delete from $[\mathbf{X}_0]_{j:}$ is

$$\ell_j - \frac{1-u_j}{u_j} \cdot r_j = \frac{u_j\ell_j - (1-u_j)r_j}{u_j} = \frac{\Delta_j}{u_j}.$$

Our strategy is to arbitrarily move mass within columns until we have deleted $\frac{\Delta_j}{u_j}$ total mass. If we denote the mass moved in column $i \in [n]$ as $\delta_{ij}$, and let $x'$ be the result after the move,

$$\langle c, x' - x \rangle \le \sum_{i \in [n]} |c_{(i,1,0)} - c_{(i,j,0)}|\delta_{ij} = \sum_{i \in [n]} ||u_j - v_i| - |u_1 - v_i||\delta_{ij} \le \sum_{i \in [n]} u_j\delta_{ij} = \Delta_j.$$

Here, the first inequality was the triangle inequality, the first equality used the definition of $c$ in (22), the second inequality used $u_1 = 0$ and the triangle inequality, and the last used $\sum_{i \in [n]} \delta_{ij} = \frac{\Delta_j}{u_j}$.

*Case 2:* $u_j \ell_j < (1 - u_j) r_j$. This case is entirely analogous; we move mass arbitrarily from row $j$ to row $m$, i.e. the last row with $u_m = 1$. The amount of mass we must move is

$$r_j - \frac{u_j}{1 - u_j} \ell_j = \frac{(1 - u_j) r_j - u_j \ell_j}{1 - u_j} = \frac{\Delta_j}{1 - u_j}.$$

Again denoting the amount of mass moved from column $i \in [n]$ as $\delta_{ij}$, the claim follows:

$$\langle c, x' - x \rangle \leq \sum_{i \in [n]} ||u_j - v_i| - |u_m - v_i|| \, \delta_{ij} \leq \sum_{i \in [n]} (1 - u_j) \delta_{ij} = \Delta_j.$$

$\square$

By iteratively applying Lemma 21, we have our marginal-preserving calibration procedure.

**Lemma 22** (Marginal-preserving calibration). *Following the notation* (24)*, given $x \in \Delta^{2mn}$ with $\|\mathbf{B}x\|_1 = \Delta$, we can compute $x'$ with $\mathbf{M}x' = \mathbf{M}x$, $\mathbf{B}x' = \mathbb{0}_m$, and $\langle c, x' - x \rangle \leq \Delta$ in $O(mn)$ time.*

*Proof.* It suffices to apply Lemma 21 to each row $i \in [m]$. All of the movement in the rows $i \in [2, m-1]$ are independent of each other, and do not affect the imbalance in the rows $i \in \{1, m\}$ when we have finished applying Lemma 21, since e.g. $u_1 \ell_1 = 0$ regardless of how much mass is moved to $[\mathbf{X}_0]_{1:}$, and a similar property holds for the $m^{\text{th}}$ row. The total change in $\langle c, x' - x \rangle$ is thus boundable by $\sum_{j \in [m]} \Delta_j \leq \Delta$, and applying Lemma 21 to each row takes $O(n)$ time. Finally, $\mathbf{M}x = \mathbf{M}x'$ follows because we only move mass within the same column, so no marginal changes. $\square$

By combining Lemma 20 with Lemma 22, we can complete our rounding procedure.

**Lemma 23.** *Let $(\mathbf{A}, b, c, \mathcal{X})$ be defined as in* (22)*,* (24)*, and let $(\widetilde{\mathbf{A}}, \tilde{b}) := (4\mathbf{A}, 4b)$. There exists* Round*, a $(\widetilde{\mathbf{A}}, \tilde{b}, 1)$-equality rounding procedure for $(\mathbf{A}, b, c, \mathcal{X})$, running in $O(mn)$ time.*

*Proof.* Throughout the proof, let $\Delta_{\mathbf{M}} := \left\| \mathbf{M}x - \frac{1}{n} \mathbb{1}_n \right\|_1$ and $\Delta_{\mathbf{B}} := \|\mathbf{B}x\|_1$, following the notation (24). We also denote the total violation by

$$\Delta := \left\| \widetilde{\mathbf{A}}x - \tilde{b} \right\|_1 = 4\Delta_{\mathbf{M}} + 4\Delta_{\mathbf{B}}.$$

We first apply Lemma 20 to $x$ to produce $\tilde{x}$ satisfying $\|x - \tilde{x}\|_1 \leq 2\Delta_{\mathbf{M}}$ and $\mathbf{M}\tilde{x} = \frac{1}{n} \mathbb{1}_n$, in $O(mn)$ time. Note that, because $\|\mathbf{B}\|_{1 \to 1} \leq 1$ since all columns of $\mathbf{B}$ are 1-sparse, we have

$$\|\mathbf{B}\tilde{x}\|_1 \leq \|\mathbf{B}x\|_1 + \|\mathbf{B}\|_{1 \to 1} \|x - \tilde{x}\|_1 \leq \Delta_{\mathbf{B}} + 2\Delta_{\mathbf{M}}.$$

Next, we apply Lemma 22 to $\tilde{x}$, resulting in $x'$ with $\mathbf{M}x' = \frac{1}{n} \mathbb{1}_n$, $\mathbf{B}x' = \mathbb{0}_m$, and $\langle c, x' - \tilde{x} \rangle \leq \Delta_{\mathbf{B}} + 2\Delta_{\mathbf{M}}$, in $O(mn)$ time. Recalling the definition (19), we have the conclusion from $\|c\|_\infty \leq 1$, so

$$c^\top (x' - x) \leq c^\top (\tilde{x} - x) + c^\top (x' - \tilde{x})$$
$$\leq \|c\|_\infty \|\tilde{x} - x\|_1 + c^\top (x' - \tilde{x}) \leq 2\Delta_{\mathbf{M}} + \Delta_{\mathbf{B}} + 2\Delta_{\mathbf{M}} \leq \Delta.$$

$\square$

We conclude by applying the solver from Corollary 2 to our resulting unconstrained linear program.

**Proposition 3.** *Let $\varepsilon \geq 0$. We can compute $x \in \mathcal{X}$, an $\varepsilon$-approximate minimizer to* (22)*, in time*

$$O\left( \frac{n \log(n)}{\varepsilon^2} \right).$$

*Further, the objective value of $x$ in* (22) *is a $2\varepsilon$-additive approximation of $\underline{\mathsf{dCE}}(\widehat{\mathcal{D}}_n)$.*

*Proof.* Observe that for $\widetilde{\mathbf{A}} = 4\mathbf{A}$, we have $\|\widetilde{\mathbf{A}}\|_{1 \to 1} \leq 8$ and $\text{nnz}(\widetilde{\mathbf{A}}) = O(mn)$, since no column is more than 2-sparse and all entries of $\mathbf{A}$ are in $[-1, 1]$. Further, recalling the definition of $U$ from (21), we have $m = O(\frac{1}{\varepsilon})$. So, Corollary 2 shows we can compute an $\varepsilon$-approximate minimizer to

$$\min_{x \in \Delta^{2mn}} c^\top x + \left\| \widetilde{\mathbf{A}}x - \tilde{b} \right\|_1$$

within the stated runtime. The rest of the proof follows using Round from Lemma 23, where we recall $|\underline{\mathsf{dCE}}(\widehat{\mathcal{D}}_n) - \underline{\mathsf{dCE}}^U(\widehat{\mathcal{D}}_n)| \leq \varepsilon$ due to our definition of $U$ and Lemma 17. $\square$

## C.4 Testing via LDTC

We now give analogs of Theorems 1 and 3, using our solver in Proposition 3.

**Theorem 4.** *Let $0 \leq \varepsilon_2 \leq \varepsilon_1 \leq 1$ satisfy $\varepsilon_1 > \varepsilon_2$, and let $n \geq C_{\text{tct}} \cdot \frac{1}{(\varepsilon_1 - \varepsilon_2)^2}$ for a universal constant $C_{\text{tct}}$. There is an algorithm $\mathcal{A}$ which solves the $(\varepsilon_1, \varepsilon_2)$-tolerant calibration testing problem with $n$ samples, which runs in time*

$$O\left(\frac{n \log(n)}{(\varepsilon_1 - \varepsilon_2)^2}\right).$$

*Proof.* Throughout the proof, let $\alpha := \varepsilon_1 - \varepsilon_2 > 0$. Consider the following algorithm.

1. For $|U| = m \geq \frac{6}{\alpha}$, sample $n \geq \frac{C_{\text{tct}}}{\alpha^2}$ samples to form an empirical distribution $\widehat{\mathcal{D}}_n$, where $C_{\text{tct}}$ is chosen so Lemma 3 guarantees $|\underline{\mathsf{dCE}}(\mathcal{D}) - \underline{\mathsf{dCE}}(\widehat{\mathcal{D}}_n)| \leq \frac{\alpha}{6}$ with probability $\geq \frac{2}{3}$.

2. Call Proposition 2 with $\varepsilon \leftarrow \frac{\alpha}{6}$ to obtain $\beta$, an $\frac{\alpha}{3}$-additive approximation to $|\underline{\mathsf{dCE}}(\widehat{\mathcal{D}}_n)|$.

3. Return "yes" if $\beta \leq \varepsilon_2 + \frac{\alpha}{2}$, and return "no" otherwise.

Conditioned on the event that $|\underline{\mathsf{dCE}}(\mathcal{D}) - \underline{\mathsf{dCE}}(\widehat{\mathcal{D}}_n)| \leq \frac{\alpha}{6}$, we show that the algorithm succeeds. First, if $\underline{\mathsf{dCE}}(\mathcal{D}) \leq \varepsilon_2$, then $\underline{\mathsf{dCE}}(\widehat{\mathcal{D}}_n) \leq \varepsilon_2 + \frac{\alpha}{6}$ by assumption, and so $\beta \leq \varepsilon_2 + \frac{\alpha}{2}$ by Proposition 2, so the tester will return "yes." Second, if $\underline{\mathsf{dCE}}(\mathcal{D}) \geq \varepsilon_1$, then $\underline{\mathsf{dCE}}(\widehat{\mathcal{D}}_n) \geq \varepsilon_1 - \frac{\alpha}{6}$ by assumption, so $\beta \geq \varepsilon_1 - \frac{\alpha}{2}$ by Proposition 2 and similarly the tester will return "no" in this case. Finally, the runtime is immediate from Proposition 3 and the definition of $\alpha$. $\square$

Theorem 4 has the following implication for (standard) calibration testing, by letting $\varepsilon_2 = 0$.

**Corollary 3.** *Let $n \in \mathbb{N}$ and let $\varepsilon_n \in (0, 1)$ be minimal such that it is information-theoretically possible to solve the $\varepsilon_n$-calibration testing problem with $n$ samples. For some $\varepsilon = \Theta(\varepsilon_n)$, there is an algorithm $\mathcal{A}$ which solves the $\varepsilon$-calibration testing problem with $n$ samples, which runs in time*

$$O\left(n^2 \log(n)\right).$$

## D Sample complexity lower bounds for calibration measures

Recent works [BN23, BGHN23a] have introduced other calibration measures (e.g. the convolved ECE and interval CE), given efficient estimation algorithms for them, and showed that they are polynomially related to the lower distance to calibration $\underline{\mathsf{dCE}}$. Therefore, an alternative approach to the (non-tolerant) testing problem for $\underline{\mathsf{dCE}}$ is by reducing it to testing problems for these measures. The main result of this section is that this approach leads to suboptimal sample complexity: the testing problems for these measures cannot be solved given only $O(\varepsilon^{-2})$ data points $\{(v_i, y_i)\}_{i \in [n]}$.

To establish this sample complexity lower bound, we construct a perfectly calibrated distribution $\mathcal{D}_0$ and a family of miscalibrated distributions $\mathcal{D}_\theta$ parameterized by $\theta$ belonging to a finite set. We use $\mathcal{D}_0^{\otimes n}$ (and $\mathcal{D}_\theta^{\otimes n}$) to denote the joint distribution of $n$ independent examples from $\mathcal{D}_0$ (and $\mathcal{D}_\theta$). In Lemma 24, we show that the total variation distance between $\mathcal{D}_0^{\otimes n}$ and the mixture $\mathbb{E}[\mathcal{D}_\theta^{\otimes n}]$ of $\mathcal{D}_\theta^{\otimes n}$ is small unless $n$ is large, and thus distinguishing them requires large sample complexity. Consequently, the testing problem for a calibration measure has large sample complexity if it assigns every $\mathcal{D}_\theta$ a large calibration error. Finally, we show every $\mathcal{D}_\theta$ indeed has large convolved ECE and interval CE, establishing sample complexity lower bounds for these measures in Theorems 5 and 6.

To construct $\mathcal{D}_0$ and $\mathcal{D}_\theta$, we consider $t$ values $\{u_i\}_{i \in [t]} \in [\frac{1}{3}, \frac{2}{3}]$ where $u_i = \frac{1}{3} + \frac{i}{3t}$ for $i \in [t]$. We will determine the value of $t \in \mathbb{N}$ later. We also define the following distribution, a perfectly calibrated distribution which is related to the miscalibrated synthetic dataset used in Section 3.

**Definition 8.** *The distribution $\mathcal{D}_0$ of $(v, y) \in [0, 1] \times \{0, 1\}$ is defined such that the marginal distribution of $v$ is uniform over $\{u_i\}_{i \in [t]}$ and $\mathbb{E}_{\mathcal{D}_0}[y|v] = v$.*

Fix $\alpha \in (0, \frac{1}{3})$. For $\theta \in \{-1, 1\}^t$, we define distribution $\mathcal{D}_\theta$ of $(v, y) \in [0, 1] \times \{0, 1\}$ such that the marginal distribution of $v$ is uniform over $\{u_i\}_{i \in [t]}$ and $\mathbb{E}_{\mathcal{D}_\theta}[y|v = u_i] = u_i + \theta_i \alpha$. In other words,

each conditional distribution given $v$ is miscalibrated by $\alpha$, but the bias takes a random direction. We now follow a standard approach by [IS03] to bound the total variation between our distributions.

**Lemma 24.** *For any $t \in \mathbb{N}$ and $\alpha \in (0, \frac{1}{3})$,*

$$d_{TV}(\mathcal{D}_0^{\otimes n}, \mathbb{E}_\theta[\mathcal{D}_\theta^{\otimes n}]) \leq \frac{1}{2}\sqrt{\exp\left(\frac{11\alpha^4 n^2}{t}\right) - 1}.$$

*Here, to construct the mixture distribution $\mathbb{E}_\theta[\mathcal{D}_\theta^{\otimes n}]$, we first draw $\theta \sim_{\text{unif.}} \{-1,1\}^t$, and then draw $n$ independent examples from $\mathcal{D}_\theta$. We denote the distribution of the $n$ examples by $\mathbb{E}_\theta[\mathcal{D}_\theta^{\otimes n}]$.*

*Proof.* By a standard inequality between the total variation distance and the $\chi^2$ distance, we have

$$d_{TV}(\mathcal{D}_0^{\otimes n}, \mathbb{E}_\theta[\mathcal{D}_\theta^{\otimes n}]) \leq \frac{1}{2}\sqrt{\chi^2(\mathbb{E}_\theta[\mathcal{D}_\theta^{\otimes n}]\|\mathcal{D}_0^{\otimes n})}. \tag{25}$$

By Ingster's method [IS03] (see also Section 3.1 of [Can22]),

$$\chi^2(\mathbb{E}_\theta[\mathcal{D}_\theta^{\otimes n}]\|\mathcal{D}_0^{\otimes n}) = \mathbb{E}_{\theta,\theta'}\left[\left(\sum_{i=1}^t \sum_{j \in \{0,1\}} \frac{\mathcal{D}_\theta(u_i,j)\mathcal{D}_{\theta'}(u_i,j)}{\mathcal{D}_0(u_i,j)}\right)^n\right] - 1, \tag{26}$$

where the expectation is over $\theta, \theta'$ drawn i.i.d. $\sim_{\text{unif.}} \{-1,1\}^t$. For every $i \in [t]$, we have

$$\frac{\mathcal{D}_\theta(u_i,1)\mathcal{D}_{\theta'}(u_i,1)}{\mathcal{D}_0(u_i,1)} = \frac{\left(\frac{u_i+\theta_i\alpha}{t}\right)\left(\frac{u_i+\theta_i'\alpha}{t}\right)}{\frac{u_i}{t}} = \frac{u_i}{t} + \frac{(\theta_i+\theta_i')\alpha}{t} + \frac{\theta_i\theta_i'\alpha^2}{u_it},$$

and similarly

$$\frac{\mathcal{D}_\theta(u_i,0)\mathcal{D}_{\theta'}(u_i,0)}{\mathcal{D}_0(u_i,0)} = \frac{\left(\frac{(1-u_i-\theta_i\alpha)}{t}\right)\left(\frac{(1-u_i-\theta_i'\alpha)}{t}\right)}{\frac{1-u_i}{t}} = \frac{1-u_i}{t} - \frac{(\theta_i-\theta_i')}{t} + \frac{\theta_i\theta_i'\alpha^2}{(1-u_i)t}.$$

Adding up the two equations, we get

$$\sum_{j \in \{0,1\}} \frac{\mathcal{D}_\theta(u_i,j)\mathcal{D}_{\theta'}(u_i,j)}{\mathcal{D}_0(u_i,j)} = \frac{1}{t} + \frac{\theta_i\theta_i'\alpha^2}{t}\left(\frac{1}{u_i} + \frac{1}{1-u_i}\right) \leq \frac{1}{t} + \frac{9\theta_i\theta_i'\alpha^2}{2t},$$

where the last inequality uses the fact that $u_i \in [\frac{1}{3}, \frac{2}{3}]$. Plugging this into (26), we get

$$\chi^2(\mathbb{E}_\theta[\mathcal{D}_\theta^{\otimes n}]\|\mathcal{D}_0^{\otimes n}) \leq \mathbb{E}_{\theta,\theta'}\left[\left(1 + \frac{9\alpha^2}{2t}\sum_{i=1}^t \theta_i\theta_i'\right)^n\right] - 1$$

$$\leq \mathbb{E}_{\theta,\theta'}\left[\exp\left(\frac{9\alpha^2 n}{2t}\sum_{i=1}^t \theta_i\theta_i'\right)\right] - 1$$

$$= \prod_{i=1}^t \mathbb{E}_{\theta,\theta'}\left[\exp\left(\frac{9\alpha^2 n}{2t}\theta_i\theta_i'\right)\right] - 1$$

$$\leq \prod_{i=1}^t \mathbb{E}_\theta\left[\exp\left(\frac{81\alpha^4 n^2}{8t^2}\theta_i^2\right)\right] - 1 \qquad \text{(by Hoeffding's lemma)}$$

$$= \exp\left(\frac{81\alpha^4 n^2}{8t}\right) - 1.$$

Plugging this into (25) completes the proof. $\qquad \square$

## D.1 Lower bound for convolved ECE

We now introduce the definition of convolved ECE from [BN23], and show that for every $\theta \in \{-1,1\}^t$, $\mathcal{D}_\theta$ has a large convolved ECE in Lemma 26. This allows us to prove our sample complexity lower bound for convolved ECE in Theorem 5, by applying Lemma 24.

**Definition 9** (Convolved ECE [BN23]). *Let $\pi_R : \mathbb{R} \to [0,1]$ be the periodic function with period $2$ satisfying $\pi_R(v) = v$ if $v \in [0,1]$, and $\pi_R(v) = 2 - v$ if $v \in [1,2]$. Consider a distribution $\mathcal{D}$ over $[0,1] \times \{0,1\}$. For $(v,y) \sim \mathcal{D}$, define random variable $\hat{v} := \pi_R(v + \eta)$, where $\eta$ is drawn independently from $\mathcal{N}(0, \sigma^2)$ for a parameter $\sigma \geq 0$. The $\sigma$-convolved ECE is defined as follows:*

$$\mathsf{cECE}_\sigma(\mathcal{D}) := \mathbb{E}|\mathbb{E}[(y-v)|\hat{v}]|,$$

*where the outer expectation is over the marginal distribution of $\hat{v}$, and the inner expectation is over the conditional distribution of $(y,v)$ given $\hat{v}$. It has been shown in [BN23] that $\mathsf{cECE}_\sigma(\mathcal{D}) \in [0,1]$ is a nonincreasing function of $\sigma \geq 0$ and there exists a unique $\sigma^* \geq 0$ satisfying $\mathsf{cECE}_{\sigma^*}(\mathcal{D}) = \sigma^*$. The convolved ECE $\mathsf{cECE}(\mathcal{D})$ is defined to be $\mathsf{cECE}_{\sigma^*}(\mathcal{D})$.*

We also mention that the following relationship is known between $\mathsf{cECE}$ and $\underline{\mathsf{dCE}}$.

**Lemma 25** (Theorem 7, [BN23]). *For any distribution $\mathcal{D}$ over $[0,1] \times \{0,1\}$, it holds that*

$$\frac{1}{2}\underline{\mathsf{dCE}}(\mathcal{D}) \leq \mathsf{cECE}(\mathcal{D}) \leq 2\sqrt{\underline{\mathsf{dCE}}(\mathcal{D})}.$$

We have the following lower bound on $\mathsf{cECE}(\mathcal{D}_\theta)$:

**Lemma 26.** *For integer $t \geq 3$, choose $\alpha = \frac{1}{t\sqrt{\ln t}}$. Then for every $\theta \in \{-1,1\}^t$,*

$$\mathsf{cECE}(\mathcal{D}_\theta) \geq \frac{1}{100t\sqrt{\ln t}}.$$

*Proof.* It suffices to show that $\mathsf{cECE}_\sigma(\mathcal{D}_\theta) \geq \frac{1}{100t\sqrt{\ln t}}$ whenever $\sigma \leq \frac{1}{100t\sqrt{\ln t}}$.

Consider $(v,y) \sim \mathcal{D}$ and $\hat{v} = \pi_R(v+\eta)$, where $\eta$ is drawn independently from $\mathcal{N}(0,\sigma^2)$. By standard Gaussian tail bounds, we have

$$\Pr\left[|\eta| \geq \frac{1}{6t}\right] \leq \frac{1}{t^2}. \tag{27}$$

Next, consider a function $\ell : [0,1] \to [t]$ such that $\ell(\hat{v}) \in \arg\min_{i \in [t]} |u_i - \hat{v}|$. Let $\mathcal{E}$ denote the event that $v = u_{\ell(\hat{v})}$. Let $\mathbb{1}_\mathcal{E}$ and $\mathbb{1}_{\neg\mathcal{E}}$ be the indicators of $\mathcal{E}$ and its complement, respectively. We have

$$
\begin{aligned}
\mathbb{E}[(y-v)\mathbb{1}_\mathcal{E} \mid \hat{v}, v] &= \mathbb{1}_\mathcal{E}\mathbb{E}[y-v \mid \hat{v}, v] && (\mathbb{1}_E \text{ is fully determined by } v \text{ and } \hat{v}) \\
&= \mathbb{1}_\mathcal{E}\mathbb{E}[y-v \mid v] && (y \text{ is independent of } \hat{v} \text{ given } v) \\
&= \mathbb{1}_\mathcal{E}\mathbb{E}[y-v \mid v = u_{\ell(\hat{v})}] = \mathbb{1}_\mathcal{E}\theta_{\ell(\hat{v})}\alpha.
\end{aligned}
$$

Taking expectation over $v$ conditioned on $\hat{v}$, we have

$$|\mathbb{E}[(y-v)\mathbb{1}_\mathcal{E} \mid \hat{v}]| = |\Pr[\mathcal{E} \mid \hat{v}]\theta_{\ell(\hat{v})}\alpha| = \Pr[\mathcal{E} \mid \hat{v}]\alpha.$$

We also have

$$\left|\mathbb{E}\left[(y-v)\mathbb{1}_{\neg\mathcal{E}} \mid \hat{v}\right]\right| \leq \mathbb{E}\left[|(y-v)\mathbb{1}_{\neg\mathcal{E}}| \mid \hat{v}\right] \leq \Pr[\neg\mathcal{E} \mid \hat{v}].$$

Therefore,

$$|\mathbb{E}[y-v \mid \hat{v}]| \geq \Pr[\mathcal{E}|\hat{v}]\alpha - \Pr[\neg\mathcal{E} \mid \hat{v}].$$

Taking expectations over $\hat{v}$, we have

$$\mathsf{cECE}_\sigma(\mathcal{D}_\theta) = \mathbb{E}[|\mathbb{E}[y-v \mid \hat{v}]|] \geq \Pr[\mathcal{E}]\alpha - \Pr[\neg\mathcal{E}]. \tag{28}$$

Whenever $\mathcal{E}$ does not occur, it must hold that $|v - \hat{v}| \geq \frac{1}{6t}$, which can only hold when $|\eta| \geq \frac{1}{6t}$. Therefore, by plugging (27) into (28), we get

$$\mathsf{cECE}_\sigma(\mathcal{D}_\theta) \geq \left(1 - \frac{1}{t^2}\right)\alpha - \frac{1}{t^2} \geq \frac{1}{100t\sqrt{\ln t}}. \qquad \square$$

**Theorem 5.** *If $\mathcal{A}$ is an $\varepsilon$-$\mathsf{cECE}$ tester with $n$ samples (Definition 5), for $\varepsilon \in (0, \frac{1}{3})$, then*

$$n = \Omega\left(\frac{1}{\varepsilon^{2.5}\ln^{0.25}(\frac{1}{\varepsilon})}\right).$$

*Proof.* Without loss of generality, assume that $\varepsilon \leq 10^{-3}$. Let $t \geq 3$ be the largest integer satisfying $\varepsilon \leq \frac{1}{100t\sqrt{\ln t}}$. We choose $\alpha = \frac{1}{t\sqrt{\ln t}}$.

By Lemma 26, we have $\mathsf{cECE}(\mathcal{D}_\theta) \geq \varepsilon$ for every $\theta \in \{-1, 1\}^t$. By the guarantee of the tester, we have

$$\mathrm{d_{TV}}(\mathcal{D}_0^{\otimes n}, \mathbb{E}_\theta[\mathcal{D}_\theta^{\otimes n}]) \geq \frac{1}{3}.$$

Combining this with Lemma 24, we get $n = \Omega(\alpha^{-2}\sqrt{t}) = \Omega(t^{2.5} \ln t)$, so the claim holds. $\qquad\square$

## D.2 Lower bound for (surrogate) interval CE

The *interval calibration error* was introduced in [BGHN23a] as a modified version of the popular *binned ECE* to obtain a polynomial relationship to the lower distance to calibration (dCE). To give an efficient estimation algorithm, [BGHN23a] considered a slight variant of the interval calibration error, called the *surrogate interval calibration error*, which preserves the polynomial relationship. Below we include the definition of the surrogate interval calibration error, and its polynomial relationship with dCE. We then establish our sample complexity lower bound (Theorem 6) for surrogate interval CE by showing that every $\mathcal{D}_\theta$ has a large surrogate interval CE (Lemma 28).

**Definition 10** ([BGHN23a])**.** *For a distribution $\mathcal{D}$ over $[0, 1] \times \{0, 1\}$ and an interval width parameter $w > 0$, the* random interval calibration error *is defined to be*

$$\mathsf{RintCE}(\mathcal{D}, w) := \mathbb{E}_r\left[\sum_{j \in \mathbb{Z}} |\mathbb{E}_{(v,y)\sim\mathcal{D}}[(y - v)\mathbb{I}(v \in I_{r,j}^w)]|\right], \qquad (29)$$

*where the outer expectation is over $r$ drawn uniformly from $[0, w)$ and $I_{r,j}^w$ is the interval $[r + j\varepsilon, r + (j + 1)\varepsilon)$. Note that although the summation is over $j \in \mathbb{Z}$, there are only finitely many $j$ that can contribute to the sum (which are the $j$ that satisfy $I_{r,j}^w \cap [0, 1] \neq \emptyset$). The* surrogate interval calibration error *is defined as follows:*

$$\mathsf{SintCE}(\mathcal{D}) := \inf_{k \in \mathbb{Z}_{\geq 0}} \left(\mathsf{RintCE}(\mathcal{D}, 2^{-k}) + 2^{-k}\right).$$

**Lemma 27** (Theorem 6.11, [BGHN23a])**.** *For any distribution $\mathcal{D}$ over $[0, 1] \times \{0, 1\}$, it holds that*

$$\underline{\mathsf{dCE}}(\mathcal{D}) \leq \mathsf{SintCE}(\mathcal{D}) \leq 6\sqrt{\underline{\mathsf{dCE}}(\mathcal{D})}.$$

**Lemma 28.** *For $t \in \mathbb{N}$, let $\alpha = \frac{1}{3t}$. Then for every $\theta \in \{-1, 1\}^t$, it holds that $\mathsf{SintCE}(\mathcal{D}_\theta) \geq \frac{1}{3t}$.*

*Proof.* It suffices to prove that $\mathsf{RintCE}(\mathcal{D}_\theta, w) \geq \frac{1}{3t}$ whenever $w < \frac{1}{3t}$, where we recall the definition (29). Fix some $r \in [0, 1]$. Every $u_i$ belongs to the interval $I_{r,j_i}^w$ for a unique $j_i \in \mathbb{Z}$. Since the interval width $w$ is smaller than the gap between $u_i$ and $u_{i'}$ for distinct $i, i'$, we have $j_i \neq j_{i'}$. Therefore,

$$\begin{aligned}
\sum_{j \in \mathbb{Z}} |\mathbb{E}_{(v,y)\sim\mathcal{D}}[(y - v)\mathbb{I}(v \in I_{r,j}^w)]| &\geq \sum_{i \in [t]} |\mathbb{E}_{(v,y)\sim\mathcal{D}}[(y - v)\mathbb{I}(v \in I_{r,j_i}^w)]| \\
&= \sum_{i \in [t]} |\mathbb{E}_{(v,y)\sim\mathcal{D}}[(y - v)\mathbb{I}(v = u_i)]| \\
&= \sum_{i \in [t]} \Pr[v = u_i]\mathbb{E}[y - v | v = u_i] \\
&= \frac{1}{3t}.
\end{aligned}$$

Plugging this into (29), we get $\mathsf{RintCE}(\mathcal{D}_\theta, w) \geq \frac{1}{3t}$. $\qquad\square$

**Theorem 6.** *If $\mathcal{A}$ is an $\varepsilon$-SintCE tester with $n$ samples (Definition 5), for $\varepsilon \in (0, \frac{1}{3})$, then*

$$n = \Omega\left(\frac{1}{\varepsilon^{2.5}}\right).$$

*Proof.* Choose $t \geq 1$ to be the largest integer satisfying $\frac{1}{3t} \geq \varepsilon$, and choose $\alpha = \frac{1}{3t}$. By Lemma 28, $\mathsf{SintCE}(\mathcal{D}_\theta) \geq \varepsilon$ for every $\theta \in \{-1, 1\}^t$. By the guarantee of the tester, we have

$$d_{\mathrm{TV}}(\mathcal{D}_0^{\otimes n}, \mathbb{E}_\theta[\mathcal{D}_\theta^{\otimes n}]) \geq \frac{1}{3}.$$

Combining this with Lemma 24, we get $n = \Omega(\alpha^{-2}\sqrt{t}) = \Omega(\varepsilon^{-2.5})$. $\qquad\square$

