# OpenReview forum: "Testing Calibration in Nearly-Linear Time"
_NeurIPS.cc/2024/Conference — NeurIPS 2024 poster_

### Official Review · Reviewer_nmkM · 2024-06-20

**Soundness:** 3
**Presentation:** 3
**Contribution:** 3
**Rating:** 7
**Confidence:** 4

**Summary:**

This paper considers the property testing of calibration, under the lower distance to calibration metric. By property testing, the authors adapt the standard definition in the TCS literature, where if the distance is at least $\epsilon$ then the algorithm will reject, and accept if the distance is 0. The main contributions are algorithms for solving the aforementioned testing problem in nearly linear time, in particular, authors show that

* The $\epsilon$-calibration testing problem could be solved in time $O(n\log^2 n)$;

* For a tolerant version of calibration testing, where the algorithm accepts if the distance $\leq \epsilon_1$ and rejects if the distance $\geq \epsilon_2$, there is an algorithm that solves it in $O((\epsilon_1-\epsilon_2)^{-2} n\log n)$ time. If one chooses $\epsilon_1-\epsilon_2=1/\sqrt n$, which is the information-theoretically smallest possible choice, then the runtime is nearly quadratic: $O(n^2 \log n)$.

The first result relies on solving the LP associated to the smooth calibration error, formulating it as a min-cost flow on a path graph adjoint by a single vertex. They show this structured LP could be solved with simple dynamic programming plus segment trees, without resorting the heavy machinery of IPMs and they empirically verify their algorithm outperforms commercial solvers (which is not very surprising, since the graph is so simple).

The second result solves a harder LP, using the rounding framework for combinatorial optimization, particularly solvers that could utilize area convexity well due to Sherman and extended by Jambulapati, Sidford and Tian. This improves upon using the state-of-the-art LP solver in a black-box way.

**Strengths:**

The theoretical results in this paper are quite interesting. This seems to be the first paper to formalize calibration as a property testing problem, and they give fast algorithms under the lower distance to calibration error. Techniques are not complicated but neat. In particular, authors show that the smooth calibration error LP could be cast into a min-cost flow problem on a simple, planar graph. Utilizing the graph structure, they develop combinatorial, dynamic programming-based approach that runs in nearly linear time. This is in drastic difference from most endeavors in algorithmic optimization community, where most works focus on developing complicated data structures and machinery to solve an IPM. This also enables authors to implement their algorithm and compares with CVXPY / commercial solvers in terms of efficiency, which is not known for most theoretically-efficient LP solvers since 2019.

**Weaknesses:**

The paper is overall well-written with solid results, however the paper format of NeurIPS is not particularly suitable for this paper. Theoretically, it also contains an algorithm for tolerant calibration testing, and a sample complexity lower bound. However, these results are only roughly conveyed in the main body of the paper. I wonder whether journals (JMLR, TMLR) or conferences such as COLT might be a better fit.

**Questions:**

A few questions:

* For smooth calibration, the runtime seems not depending on $\epsilon$. Is that because $\epsilon$ is chosen as $1/\sqrt n$, or smooth calibration is a constant factor approximation to lower dCE hence $\epsilon$ is a constant? I think it's important to clarify what role does $\epsilon$ play in the smooth calibration program.

* Do you think the method of transforming smooth calibration to computing a min-cost flow on a path adjoint with a vertex could be generalized to other programs where the constraints are encoding Lipschitzness among the data points? Or is the definition of vector $b$ crucial here?

A few comments:

* For LP solvers, consider citing the state-of-the-art result by Jiang, Song, Weinstein and Zhang, STOC'21.

**Limitations:**

Yes.

---

> ### Author Rebuttal · Authors · 2024-08-06
>
> Thank you for your positive review and careful reading. We are also encouraged that we were able to develop a practical solver for our LP of interest by using its structure, which uses less sophisticated tools than the recent LP solvers from the TCS community.
>
> Theorem 1’s runtime does not depend on $\epsilon$ because our method exactly solves the empirical smCE problem. This is made possible by designing an exact solver for the corresponding LP, a structured min-cost flow instance. The $\epsilon \approx 1/\sqrt{n}$ error in the tester is entirely due to the inaccuracy from the sampling process, see Lemma 3. On the other hand, the LP solver we designed in Theorem 2 is only approximate, leading to two sources of error in the final guarantee. We will clarify these distinctions in our revision.
>
> Thank you for your insightful question about generalizations of our approach in Appendix B.2. Indeed, we are optimistic that our approach generalizes to other settings; in follow up work, we extended our algorithm to handle structured quadratic programs with Lipschitz constraints, a common problem formulation in isotonic regression (see e.g. the LPAV subproblem in “Efficient Learning of Generalized Linear and Single Index Models with Isotonic Regression”).
>
> We believe you can convert Lipschhtiz encoding constraints for linear programs to a min-cost flow problem problem with the structure you mentioned. The value of $b$ is not important here; it can range $[-1,1]$. This question is exactly what we are considering for the next step. We are working on a problem called LPAV, which involves solving quadratic programs with Lipschtiz constraints, and we are also looking for other problems with Lipschtiz constraints.
>
> We will add a reference to [JSWZ21] in our revision; thank you for the reminder.

---

> > ### Comment · Reviewer_nmkM · 2024-08-07
> >
> > I thank the authors for addressing my questions. I'll keep my score as is.

---

### Official Review · Reviewer_rwaD · 2024-07-08

**Soundness:** 3
**Presentation:** 3
**Contribution:** 3
**Rating:** 7
**Confidence:** 3

**Summary:**

This paper introduces a property testing formulation of verifying calibration, along with efficient algorithms for solving this problem as well as a relaxed/tolerant version. The empirical results support the theory and justify the efficacy of the proposed algorithms.

**Strengths:**

**Originality:** The calibration testing formulation is new (although a natural extension of prior work [1]), and consequently the algorithm for solving the problem is also new.

**Quality:** The paper is technically sound. I did not check the main proof details in the appendix, but I did check the in-lined proofs in the main body and I saw no issues.

**Clarity:** Overall, the paper is straightforward to read and the necessary background is properly introduced.

**Significance:** There has been a significant amount of recent theoretical work on how to properly measure calibration, and this paper makes a nontrivial contribution both in terms of perspective and methodology.

[1] https://arxiv.org/abs/2203.01850

**Weaknesses:**

Overall, I find the paper strong in what it sets out to achieve (and thus recommend accept); the main weakness in my view is proper contextualization and how to apply the ideas to larger scale calibration settings.

In particular, while the authors emphasize how different calibration metrics such as smoothECE perform worse for property testing, it would be interesting to have more concrete applications to model comparison in practice. For example, are there instances where comparing models using directly computed calibration metric values (i.e. ranking by smoothECE) leads to spurious rankings that would have been detected by testing with dCE? The results in Table 1 suggest this should be the case, but I think more clearly demonstrating how the proposed framework can be used as a better comparison method in practical settings would be very useful. Relatedly, I have some questions regarding the DenseNet experimental setup that I outline below under Questions.

Minor Comments:
- There seems to be some mixing up of directed/undirected throughout Lemma 2.

**Questions:**

- The DenseNet experiments rely on heavily subsampling the test data; this should not be necessary for the proposed approach correct? It seems to me that even the compared-to approaches should be fine on CIFAR-10-scale data without subsampling? Additionally, the subsampling setup seems somewhat arbitrary - would it be possible to provide more justification on the chosen sample size, the number of samples, etc.?

- What is referred to as ConvECE in the paper was actually introduced as SmoothECE, correct? It would be useful to clarify this.

**Limitations:**

Limitations were discussed throughout the paper, although there is no separate limitations section.

---

> ### Author Rebuttal · Authors · 2024-08-06
>
> Thank you for your encouraging feedback.
>
> We agree with you that applications of our results to model selection in more concrete settings is an exciting potential implication of our paper, and believe that this is a natural further direction to take our techniques. Because the algorithmic theory of efficient calibration testing was relatively less-studied prior to our work, we chose to make this topic the focus of our paper. As you suggested, in light of the difficulties in practice encountered when using binned ECE variants (Line 53), it is very interesting to see if we can provide practical criteria for settings where dCE succeeds at measuring calibration but binned alternatives such as smoothECE are less reliable. We believe this question is highly-related to the questions we study, but outside our intended scope, and defer a more thorough investigation to future work.
>
> Thanks for pointing out consistency issues in Lemma 2. We will do a thorough pass to clarify this in our revision; it is intended that $G$ (as in the lemma statement) is undirected, and $\widetilde{G}$ (as in the proof) is a directed graph, constructed from splitting edges in $G$.
>
> Our method indeed works for the dataset without subsampling. We aimed to make our experimental design decisions consistent with those used in the paper “On Calibration of Modern Neural Networks,” to show that smooth calibration is a reliable calibration measure. However, our method works fine without subsampling; indeed, our experiments suggest that our algorithm should scale better to large dataset sizes than alternative LP-based approaches.
>
> We renamed smECE to convECE because smECE and smCE are different, and we thought this similar naming might confuse readers, as smCE is central to our paper. We will clarify this point.

---

> > ### Comment · Reviewer_rwaD · 2024-08-10
> >
> > Thank you for your response, and clearing up my questions. As my score was already on the acceptance side, I will maintain it.

---

### Official Review · Reviewer_F42r · 2024-07-11

**Soundness:** 3
**Presentation:** 3
**Contribution:** 3
**Rating:** 6
**Confidence:** 4

**Summary:**

The paper studies the problem of *calibration testing*.
Here, we are given a distribution $D$ over outcomes and the goal
is to decide if the distribution is calibrated; specifically,
the property testing problem they formulate distinguishes
between perfectly calibrated distributions and those that
are $\epsilon$-far from calibrated.
A distribution $D$ over prediction-outcomes
$[0, 1] \times \{0, 1\}$ is said to be perfectly calibrated
if $E_{(v, y) \sim D}[y \mid v] = v$. The "distance to calibration" defined here
is based on [BGHN23a] which is intuitively an optimal transport
definition;
they set $dCE(D)$ to be minimum of $E_{(u, v, y) \sim \Pi}[|u - v|]$
across all $\pi$ such that $(v, y)$ has marginal distribution $D$
and $(u, y)$ is a perfectly calibrated distribution.

The authors design a dynamic-algorithm based solver for the problem.
A key insight here is a novel algorithm
for calculating the smooth calibration error
by reformulating it as a min-cost flow linear program on specific graph.
Utilizing the properties of the graph (which is a union of a star and a path),
the authors obtain a dynamic programming based algorithm which has an update time of $O(n \log^2(n))$.
This improves on the existing bounds that are $O(n^{\omega})$ where $\omega$ is the matrix multiplication exponent.

**Strengths:**

Strengths:
- The paper is studies an interesting problem, is generally well-written, and makes important contributions.
- The techniques are interesting, in particular the algorithm for calculating
  the Smooth calibration error is of independent interest.

**Weaknesses:**

Weakness:
- Framing: it appears to me that the main contribution of the paper is a new algorithm for calculating smCE.
  Indeed, the main "property testing" part of the paper (excluding the appendix) seems to be Lemma 3 which is from
  prior work. While the implications for property testing are interesting, it seems to me that they are
  mostly known. Lemma 5 also seems standard for property testing and seems like it holds for a much more general
  class of testing problems, rather than being specific to calibration.
- Proof of Lemma 2 should be better explained (especially since it is the central insight in the paper).
  For instance, what does each variable of the program correspond to in the flow?
  How does optimization problem (7) intuitively correspond to the min-flow problem of the graph specified
  in Lemma 2? What does the constraint $B^Tf=d$ mean here?

**Questions:**

See weaknesses above.
In addition, I think adding further motivation for the problem of testing calibration could improve the paper.

- The linear program (2) seems to be already provided in [BGHN23a]; see Theorem 7.14. If this is the case, it should be noted more clearly.
- The theorem statements imply that $\epsilon_n=\Theta(n^{-1/2})$ is the range in which
  the problem can be information theoretically solved. Lemma 5 provides a lower bound;
  the upper bounds is Lemma 3; this should be noted in the intro.

---

> ### Author Rebuttal · Authors · 2024-08-06
>
> Thank you for your encouraging review. We are glad that you found the problem we study interesting, and that our paper makes important contributions to it.
>
> We apologize for any confusion in framing. We agree that our Theorem 1 is really just a fast solver for the empirical smCE. However, we presented our paper through the lens of calibration testing, a problem of significant practical and theoretical interest, because it is the main overall problem we address. The reason we designed a nearly-linear time smCE algorithm is exactly because of its implications for calibration testing (building upon results from [BGHN23a], who relied on black-box LP solvers and did not give a fast algorithm). We also provide a suite of additional results on calibration testing, e.g. a tolerant tester in Appendix C which does not go through smCE, and sample complexity lower bounds in Appendix D which use other calibration metrics. We will clarify the organization of the paper so that the framing is more clear.
>
> Re: Lemma 2, our claim in (7) is that for any matrix $B$ and vectors $d, c$, the linear program $\max d^T x$ s.t. $Bx \leq c$ has an equivalent dual $\min c^T f$ s.t. $f \geq 0$, $B^T f = d$. This statement uses nothing about the graph structure of our LP. We chose the letters $f$ for the dual LP variable (the Lagrange multipliers for the primal Lipschitz constraints $Bx \leq c$) because it happens to be that our constraint matrix does have graph structure, so $f$ is naturally interpreted as a flow. In particular, our matrix $B$ is the adjacency matrix of a graph. In particular, the way we defined $B$ (in 6) means every row of $B$ has one $1$ and one $-1$. Thus, $B^T f = d$ is just specifying the net amount of flow going into each vertex (d is the vertex demands); each entry of $B^T f$ sums all the flows leaving a vertex, and subtracts all the flows coming in. We will make sure to add more explanation of this intuition in the revision; you are right that it is rather terse at present.
>
> You are correct that the linear program formulations of smooth calibration (2) and lower distance to calibration (Lemma 18) are both from [BGHN23a]. Our main contribution is a new efficient algorithm for exactly solving (2) via solving a structured min-cost flow problem. We also provide a faster approximation algorithm for the LP formulation in Lemma 18. We will make sure it is more clearly stated that both the LPs corresponding to smCE and LDTC are from [BGHN23a].
>
> We will also make sure to add a paragraph in the revision pointing out the sample complexity bounds you mention, and explaining their implications more clearly.
>
> Thank you for your careful reading and valuable feedback once again; we hope this discussion was clarifying, and elevates the merits of our paper in your view.

---

> > ### Comment · Reviewer_F42r · 2024-08-11
> >
> > Thank you for the response; I have no further questions.

---

### Official Review · Reviewer_q4wr · 2024-07-13

**Soundness:** 2
**Presentation:** 2
**Contribution:** 2
**Rating:** 5
**Confidence:** 2

**Summary:**

This paper studies testing the calibration of predictors through joint distributions and contributes efficient methods using appropriate measures.

**Strengths:**

Originality:
There are new methods.
The work can be considered a novel combination of well-known techniques.
It is clarified how this work differs from previous contributions.
Related work appears to be adequately cited.

Quality:
The submission appears to be technically sound.
The claims are rather well supported with experimental results.
The methods used are appropriate.
This is a rather complete piece of work.

Clarity:
The submission is somewhat clearly written.

Significance:
The results seem important.
Other researchers and practitioners will possibly use the ideas or build on them, owing to the code availability.
The submission seems to address a difficult task in a better way than previous works.
The work advances the state of the art as demonstrated via run-times.
The work provides provide unique theoretical and experimental approaches with demonstrated run-time gains.

**Weaknesses:**

Quality:
The authors are not careful (and possibly honest) about evaluating their work, specifically weaknesses and limitations.

Clarity:
The work is not well organized, many parts (deferred to appendix) need to be in the paper.
The work informs the reader in rather superficial level, with regards to exact implementations.

**Questions:**

Major Questions:
- Definition 1, clarify the empirical forms of dCE and ECE and the benefits dCE brings about.
- "Page 3 Line 83" and Theorem 1 disagree about the information-theoretic possibility.
- Page 8 Line 288: it is unclear how Lemma 4 is achieved, more explanations are needed.

Minor Questions:
- Page 5 Line 168: why 2(n 2) constraints?
- Page 5 Line 175: the exact use of triangle inequality is needed here.
- In (5), why did it switch from max to min?

Suggestions:
- Clarify if Theorem 1 solves with smCE and Theorem 2 solves with LDTC. It is not clear why Theorem 2 suffers additional complexity.
- Deferred definitions (4 and 5) seem central and need to be included in the paper.

**Limitations:**

The authors spread the limitations throughout the paper, so a summary section is suggested, possibly in the form of a conclusion.

---

> ### Author Rebuttal · Authors · 2024-08-06
>
> We thank the reviewer for your careful reading – we are glad that you found our work to be complete, and that you found the results important.
>
> We were confused by the reviewer’s comment about the lack of care or honesty in evaluating our work – we take such concerns very seriously, and would very much like to address any potential shortcomings that the reviewer found. We aimed to provide a thorough comparison between our work and prior calibration testers (see e.g., Section 1.3), and stated all runtimes and error rates explicitly for our algorithms. We would like to request: could the reviewer be more specific about what specific sections they found lacking? In particular, this comment did not seem to be brought up again in the reviewer’s more detailed questions.
>
> Similarly, we regret that the reviewer found our organization lacking. We did our best to communicate the main takeaways from our paper, within the 9-page space constraints of the conference format, but will do our best to keep our main body more self-contained in the revision, e.g. move Definitions 4 and 5 to the body. Thank you for this suggestion.
>
> We now address your more specific questions.
>
> The empirical form of ECE and dCE are both the standard definitions (Lines 47, 63), but where the distribution D is the uniform distribution over samples (the empirical distribution). We will clarify this in our revision. In terms of benefits of dCE, the empirical ECE definition is meaningless for continuous distributions; simple counterexamples show that the empirical ECE cannot nontrivially approximate the ECE, e.g. $y$ is uniform in $\{0, 1\}$ and $v$ is independent and uniform in $[0.49, 0.51]$. We will include this counterexample in our paper, to augment Lines 48-50. In practice, binned variants of the empirical ECE are used instead (with the bin size as a hyperparameter); as discussed in Line 53, this has repeatedly been reported to cause stability issues in practice. Conversely, the empirical dCE has been observed in both theory and practice to not suffer from such stability issues, and also has no binning hyperparameter tuning required.
>
> Line 83 is a sample complexity lower bound which states we need at least $c/\epsilon^2$ samples from some constant $c$. Our Theorem 1 uses $c’/\epsilon^2$ samples for some $c’ > c$, so there is no contradiction. Note that a sample size can be simultaneously $\Omega(1/\epsilon^2)$ and $O(1/\epsilon^2)$.
>
> Lemma 4 is Theorem 7.3 in the prior work [BGHN23a]. To avoid redundancy, we refer to [BGHN23a] for the full proof, but at a high level, the side $\text{smCE}<=2\text{LDTC}$ follows from the definitions and $\frac 1 2 \text{LDTC} \le \text{smCE}$ is by LP duality. We actually prove a strengthening of this lemma holds in our Lemma 13. We will make sure this explanation is in the revision.
>
> Re: Line 168, here, we are referring to the linear constraints, so each absolute value corresponds to $2$ linear constraints. We have $\binom{n}{2}$ absolute value constraints between coordinate pairs, and therefore $2\binom{n}{2}$ when they are converted to linear constraints.
>
> Re: Line 175, we showed the exact use of the triangle inequality in the proof of Lemma 1 (Appendix B). We will add an appropriate forward reference to this.
>
> Re: (5), this was a typo, good catch. We can redefine $b := -b$, as the maximizing argument for $f$ is the minimizing argument for $-f$, and our algorithm uses no structure of $b$. We will clarify this.
>
> Theorem 2 suffers from additional complexity because it solves a more complicated problem of $\text{LDTC}$, rather than $\text{smCE}$. The constraints in $\text{LDTC}$ do not correspond to a graph-structured linear program (e.g. min-cost flow), and therefore we are not able to give a combinatorial algorithm, relying on a new custom LP solver we develop. However, Theorem 2 is able to achieve stronger testing guarantees (i.e. it applies to tolerant testing).
>
> We hope these responses were clarifying, and that they elevate the merit of our paper in your eyes. Thank you again for your reviewing efforts.

---

> > ### Comment · Reviewer_q4wr · 2024-08-10
> >
> > I have read and considered the author response.
> >
> > To answer your question, I would still suggest a conclusion section that also summarizes the weaknesses and limitations.
> >
> > I believe your work has merit and the organizational issues are outweighed. For now, I am keeping my score.

---

### Decision · Program_Chairs · 2024-09-25

**Decision:**

Accept (poster)

**Comment:**

This paper proposes a new method to test calibration, based on several types of optimization approaches. All reviewers were in agreement that this is a rigorous and well executed paper.